

# ANALYSIS OF SUMMER O₃ IN THE MADRID AIR BASIN WITH THE LOTOS-EUROS CHEMICAL TRANSPORT MODEL

Miguel Escudero[1,2], Arjo Segers[2], Richard Kranenburg[2], Xavier Querol[3],
Andrés Alastuey[3], Rafael Borge[4], David de la Paz[4], Gotzon Gangoiti[5], and
Martijn Schaap[2]

[1]Centro Universitario de la Defensa (CUD) de Zaragoza, Academia General Militar, Ctra. de Huesca s/n,
E-50090 Zaragoza, SPAIN
[2]TNO, P.O. Box 80015, 3584 TA, Utrecht, THE NETHERLANDS
[3]Institute of Environmental Assessment and Water Research (IDAEA-CSIC), Jordi Girona 18-26, E-08034
Barcelona, SPAIN
[4]Department of Chemical & Environmental Engineering, Technical University of Madrid (UPM), c/José
Gutiérrez Abascal 2, E-28006 Madrid, SPAIN
[5]Escuela Técnica Superior Ingeniería de Bilbao, Departamento Ingeniería Química y del Medio Ambiente,
Universidad del País Vasco UPV/EHU, Urkixo Zumarkalea, s/n, Bilbao, E-48013, SPAIN

**Correspondence:** Miguel Escudero (mescu@unizar.es)

**Abstract.** Tropospheric O₃ remains a major air-quality issue in the Mediterranean region. The combination of large anthropogenic emissions of precursors, transboundary contributions, a warm and dry aestival climate and topographical features results in severe cases of photochemical pollution. Chemical transport models (CTMs) are essential tools for studying O₃ dynamics and for assessing mitigation measures but they need

5    to be evaluated specifically for each air basin. In this study, we present an optimisation of the LOTOS-EUROS CTM for the Madrid air basin. Five configurations using different meteorological datasets (from the European Centre for Medium Weather Forecast (ECMWF) and Weather Research and Forecasting (WRF)), horizontal resolution and number of vertical levels were compared for July 2016. LOTOS-EUROS responded satisfactorily in the five configurations reproducing observations of surface O₃ with notable correlation and

10    reduced bias and errors. However, the best-fit simulations for surface O₃ were obtained by increasing spatial resolution and using a large number of vertical levels to reproduce vertical transport phenomena and the formation of reservoir layers. Using the optimal configuration obtained in the evaluation, three characteristic



events have been described: recirculation (REC) episodes and northern and southern advection (NAD and SAD, respectively ) events. REC events were found to produce the highest $O_3$ due to the reduced ventilation associated with low wind speeds and the contribution of reservoir layers formed by vertical transport of $O_3$ formed near the surface in the previous days of the event. NAD events, usually associated with higher wind

speeds , present the lowest ground-level $O_3$ concentrations in the region. During SAD episodes, external contributions along with low wind speeds allow $O_3$ to increase considerably, but not as much as in REC events because steady southerly winds disperse local emissions and hinder the formation of reservoir layers.

*Copyright statement.* TEXT

## 1 Introduction

Ozone ($O_3$) is formed in the troposphere by the interaction of gaseous precursors like nitrogen oxides ($NO_x$) and volatile organic compounds (VOCs) in the presence of sunlight. Much attention has been given to these secondary air pollutants in the last decades due to the variety of negative effects on health, ecosystems, crops, climate and materials associated with it (see review by Monks et al. (2015) and references therein).

   The oxidative effect of $O_3$ generates inflammation of airways. Increases in morbidity and mortality and

chronic alterations of the cardiovascular and cerebrovascular systems have also been associated with exposure to $O_3$ (WHO, 2006, 2013a, b). Tropospheric $O_3$ is also harmful for vegetation, generating leaf symptoms, reduced growth, senescence, defoliation and reducing crop productivity (Paoletti, 2006; WGE, 2013). Damage to construction materials like plastics, surface coatings and rubber due to $O_3$ has been documented (Lee et al., 1996; Screpanti and Marco, 2009). Moreover, $O_3$ in the troposphere acts as a greenhouse gas

with positive global radiative forcing (IPCC, 2013).

   It is estimated that 98% of the urban population in Europe in 2016 were exposed to excessive concentrations of tropospheric $O_3$ according to the World Health Organization (WHO) guidelines values, a steady proportion since 2000 (EEA, 2018). However, it is the Mediterranean basin where the most acute episodes are registered (Millán et al., 1997, 2000; Sicard et al., 2013; Querol et al., 2016). In the Iberian Peninsula

(IP), located at the Western Mediterranean Basin, the intense solar radiation, high temperatures and lack of



precipitation in spring and summer, associated with persistent anticyclonic conditions, favour the formation of $O_3$ in the area and the accumulation in rural and suburban regions (Escudero et al., 2014, 2016; Querol et al., 2016; Massagué et al., 2019). The emissions of precursors from anthropogenic sources in the Mediterranean basin and the surrounding regions are considerable, especially in some densely populated areas. In

addition to that, the amount of biogenic VOCs emitted in southern Europe is considerably higher than in central and northern Europe (Seco et al., 2011). Moreover, during the frequent biomass burning episodes in summer, air-quality problems associated with tropospheric $O_3$ are aggravated (Tressol et al., 2008). In particular, the complex orography of the IP with mountain ranges running parallel to the coast intersected by river basins that penetrate towards the inner continental areas and elevated plateaus in the centre of the peninsula,

help air masses to recirculate and age under the influence of sea and mountain breezes that develop when synoptic circulation is inhibited by the presence of the Azores high (Millán et al., 2000; Gangoiti et al., 2001; Valverde et al., 2016; Querol et al., 2017). Previous studies also suggest that local strategies designed to meet $NO_2$ ambient air-quality standards may have caused an increase of urban $O_3$ that, in turn, causes an increase in oxidative capacity of Madrid's atmosphere by increasing OH and $NO_3$ radicals (Saiz-López et al., 2017).

In recent years, several comprehensive summer campaigns with intensive measurements of surface and vertical profiles of $O_3$ concentrations and its precursors have been undertaken near the two main conurbations in Spain: Barcelona (2015, 2017 and 2018) and Madrid (2016) (Querol et al., 2017, 2018; Reche et al., 2018; Carnerero et al., 2018). The main objective of these campaigns was to interpret the phenomenology of high $O_3$ and ultrafine particles' episodes in Spain.

Another essential objective of retrieving data from these intensive campaigns is related to the validation and optimisation of chemical transport models (CTMs). These models constitute an essential tool for analysing $O_3$ behaviour with high spatial and time resolution, providing air-quality forecasts and supporting the design of policies. This includes the study the $NO_x$–VOC sensitivity (Sillman, 1999; Sillman and West, 2009) essential for proposing and evaluating potential mitigation measures. Regional CTMs have been used

to investigate $O_3$ pollution in Spain in several studies. Most of these studies aimed to describe short-term (rarely exceeding 5 days) pollution events (Toll and Baldasano, 2000; Jiménez et al., 2005; San José et al., 2005; Jiménez et al., 2006; Carvalho et al., 2006; Valverde et al., 2016; Pay et al., 2018) and, in some cases, to discuss the effectiveness of potential mitigation strategies (Palacios et al., 2002; Soret et al., 2014). Despite these efforts, work is still needed to evaluate the impact of changes in the vertical configuration of CTMs,



especially in the Mediterranean region where the atmospheric dynamics in summer is characterised by complex recirculation processes with effective vertical exchange (Millán et al., 1997, 2000; Gangoiti et al., 2001; Borge et al., 2010; Querol et al., 2017, 2018). The lack of an appropriate representation of the vertical variability of $O_3$ has been recognised as one of the shortcomings of the CTMs and in consequence a major

challenge in the future development of the models (Hess and Zbinden, 2013; Monks et al., 2015). Moreover, it is strongly recommended to combine modelling with observations because this will bring knowledge from both sources together and permit adequate evaluation procedures of the model outputs (Canepa and Builtjes, 2017).

   Making use of the results on the $O_3$ episodes phenomenology from the aforementioned field campaign

in Madrid in July 2016, we were able to assess and optimise LOTOS-EUROS CTM v2.0 (Manders et al., 2017) for simulating $O_3$ in this region. Five configurations combining different meteorological input data and vertical structures were employed after identifying these two aspects as key factors for the capability of the model for reproducing $O_3$ levels. We simulated the entire month of July 2016 in accordance with the experimental campaign with a spin-up period of 24 h. The aim of this comparison was to elucidate the

optimal configurations for operating with LOTOS-EUROS in the region but also to identify relevant factors to set up other CTMs used in this region.

   Moreover, employing the optimal configuration of the modelling system, we discuss the phenomenology of tropospheric $O_3$ in the Madrid air basin (MAB) for the study period. This was done by analysing simulated fields of meteorological variables and pollutants with special emphasis on the vertical variability to test the

importance of the up– down transport of $O_3$ in the region.

## 2    DATA AND METHODS

### 2.1    Study area

The Madrid Metropolitan Area (MMA) is a densely populated area with more than 5 million inhabitants. According to Salvador et al. (2015) and Borge et al. (2014), the main sources of pollutants in the region

are road traffic, residential heating (which maximize their emissions in winter), a busy airport and minor contributions from industry.





The MMA is located in the centre of the MAB and lies on an elevated plateau (∼700 m above sea level (a.s.l.)) in the middle of the IP (Figure 1). The climate in the area is continental Mediterranean with warm and dry summers and cold and also dry winters. The main orographic features surrounding the basin are, around 120 km to the south of the MMA, the Toledo Mountains (altitudes up to 1600 m a.s.l.) with an E–W axis and the Guadarrama range (maximum heights of 2400 m a.s.l.) which runs diagonally from SW to NE, 50 km to the west and north of the MMA. The Guadarrama range is part of the Central System that extends until the Ebro valley and, together with the western flank of the Iberian range delimits a channel to the NE along the Henares valley (Figure 1). As a result of this configuration, the circulation in the Madrid basin shows a dominant SW–NE direction (Plaza et al., 1997). Under low-gradient synoptic conditions, the combination of the strong convective conditions and the blocking effect of the mountain ranges induces an important vertical development of the boundary layer and mesoscale recirculation . During the night, north-easterly winds prevail over the basin and, after dawn, the eastern slopes of the Guadarrama range are progressively warmed up causing a clockwise turning of wind to an E and S during the day finalising with an SW component in the late afternoon. The drainage flows at night-time re-establish the north-easterlies.

## 2.2 The LOTOS-EUROS model

The 3D CTM LOTOS-EUROS v2.0 and its previous versions have been extensively used in the past for air-quality studies, including $NO_x$ (Schaap et al., 2013; Vlemmix et al., 2015), $SO_2$ (Barbu et al., 2009) and particulate matter (PM) (Schaap et al., 2004; Manders et al., 2009; Timmermans et al., 2017). In particular, tropospheric $O_3$ has been the scientific target in different studies carried out with successive versions of LOTOS-EUROS. It has been employed in health-related studies (van Zelm et al., 2008) and, more recently, Beltman et al. (2013) applied LOTOS-EUROS to simulate the response of tropospheric $O_3$ in Europe to a 5% shift from crop- and grassland into poplar plantations used for biomass production, while Hendriks et al. (2016) tested the response to a decarbonisation scenario in the continent. Although LOTOS-EUROS has been generally employed in a continental domain (mainly in Europe), a sensitivity study to regional changes in emissions in three areas of Europe (Poland, the Po valley and Flanders) was also performed by Thunis et al. (2015). In addition, LOTOS-EUROS has also been used in a number of intercomparison studies with other CTMs for the simulation of $O_3$ (Hass et al., 1997; van Loon et al., 2007; Cuvelier et al., 2007; Vautard et al., 2007; Solazzo et al., 2012; Im et al., 2015) showing a satisfactory performance. Fi-



nally, regarding air-quality predictions, LOTOS-EUROS participates in the CAMS (Copernicus Atmosphere Monitoring Service) ensemble (Curier et al., 2012), which offers operational forecasts for $NO_2$, $O_3$ and PM.

## 2.3 Model experimental design

A detailed description of the 2.0 version of LOTOS-EUROS can be found in its reference guide (Manders et al., 2016) where all technical issues (processes, schemes, etc.) are described and referenced (accessible at www.lotos-euros.nl). In this section, we provide a brief description focusing on the most relevant aspects for this study.

Initial sensitivity studies were performed with the base configuration (configured similar to the operational forecasts that are part of the CAMS regional ensemble as presented in Marécal et al. (2015)) to test the response of the model to changes in the deposition velocity of $O_3$ because night-time dry deposition has been suggested as a factor that could strongly influence the ability of CTMs to simulate tropospheric $O_3$ (Stevenson et al., 2006; Monks et al., 2015). The results (not shown here) reflected a minimal effect of this parameter on $O_3$ concentrations in the chosen domain and period so deepening in this direction was discarded.

As shown in Table 1, two major aspects were modified in the set of five configurations: the meteorological input data and the vertical structure of the model. We fed LOTOS-EUROS with operational data from the reanalysis of the ECMWF model (Flemming et al., 2009) retrieved with a spatial resolution of 7 x 7 $km^2$. A second meteorological gridded dataset was obtained with the WRF model (Skamarock et al., 2008) with a resolution of 1 x 1 $km^2$ over a square domain of approximately 220 km of side centred on the city of Madrid (Figure 1). Data from WRF simulations with similar configurations have been previously used to drive air-quality simulations over the IP and, in particular, in the Madrid area (Borge et al., 2008, 2014). In this case, the WRF model was run on a three -nested-domain configuration as shown in Figure S1. Additional information about the WRF model configuration is provided in Table S1.

For the vertical structure, we compared the standard five-level mixed-layer configuration (Manders et al., 2017) with a hybrid-layer multilevel scheme. This version uses the lowest 70 layers of the 137 hybrid sigma-pressure layers used by ECMWF for the operational meteorological forecasts in 2016. In such a vertical co-ordinate system, model layers are defined by pressure boundaries that follow surface pressure at lower altitudes but slowly evolve into fixed pressure levels in the stratosphere (Eckermann, 2009).


Finally, the MACC III emission inventory has been used for all set-ups and initial and boundary concentrations were taken from global simulations produced by and used in CAMS services, as described in Marécal et al. (2015). These include concentrations of the most important trace gases and aerosols.

|  | ECMWF_5 | ECMWF_70 | ECMWF_HR_70 | WRF_5 | WRF_70 |
|---|---|---|---|---|---|
| Meteorological model spatial resolution | 7 x 7 km$^2$ | | | 1 x 1 km$^2$ | |
| Vertical structure | Mixed-layer (5 levels) | Hybrid-layer (70 levels) | | Mixed-layer (5 levels) | Hybrid-layer (70 levels) |
| LOTOS-EUROS Spatial resolution | 25 x 25 km$^2$ | | 10 x 10 km$^2$ | 3 x 3 km$^2$ | |
| Emission inventory | MACC III | | | | |

**Table 1.** Summary of the LOTOS-EUROS v2.0 model runs and settings performed for this work.

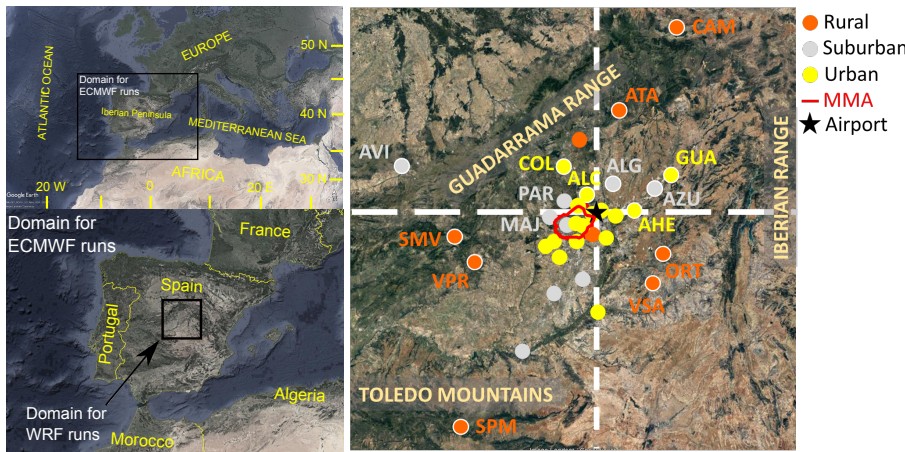

**Figure 1.** Situation maps of the Iberian Peninsula and the MAB with the indication of the domains used for the simulations with LOTOS-EUROS. In the map located on the right side, the selected set of monitoring stations used for illustrating O$_3$ variability in different sectors of the region is shown. Stations' codes: Alcalá de Henares-AHE, Alcobendas-ALC, Avila-AVI, El Atazar-ATA, Azuqueca de Henares-AZU, Campisábalos-CAM, Colmenar Viejo-COL, Guadalajara-GUA, Majadahonda-MAJ, Orusco de Tajuña-ORT, San Martín de Valdeiglesias-SMV, San Pablo de los Montes-SPM, El Pardo-PAR, Villa del Prado-VPR, Villarejo de Salvanés-VSA. The white dashed lines indicate the E–W and N–S cross-sections presented in figures from section 3.2.



## 2.4 Monitoring data

Hourly $O_3$ data for the simulation period (July 2016) were collected from 35 air-quality monitoring sites (17 urban, 9 suburban and 9 rural) located in the MAB (Table 2 and Figure 1) and its boundary region. Traffic stations were in general discarded for the model evaluation due to their limited spatial representativity al-

though four traffic sites were also employed. In spite of belonging to different air-quality networks (Table 2), all the $O_3$ monitors are based on ultraviolet photometry according to EN 14625:2012, which is the reference technique for automatic monitoring of $O_3$ established in the European Directive 2008/50/EC.

Several previous studies used modelling techniques to analyse intense short-term $O_3$ episodes (Palacios et al., 2002; San José et al., 2005), and, more specifically, to evaluate the impact of specific environmen-

tal policies in the Madrid region (Soret et al., 2014) or the influence of sectoral emissions (Borge et al., 2014; Valverde et al., 2016; de la Paz et al., 2016; Pay et al., 2018). The present work contributes with an evaluation and optimisation of LOTOS-EUROS CTM in the region incorporating a high degree of vertical resolution. The extended simulation period of this study (one month) is suitable for characterising the typical $O_3$ episodes occurring in the area in summer.

## 3 RESULTS

### 3.1 Model performance

#### 3.1.1 Ground-level $O_3$

The results from the outputs from the five runs were assessed following the suggestions provided by Borrego et al. (2008) on the evaluation of the quality of model simulations. These authors stated that Pearson's

correlation coefficient ($r$), the fractional bias (FB) and the normalised mean square error (NMSE) were the most relevant statistical parameters to be analysed for $O_3$ simulations with CTM. Figure 2 shows a coloured grid indicating the values of $r$, FB and NMSE obtained in the comparison between the observations from background monitoring stations and the simulated $O_3$. The colour scale is independent for each parameter and serves as an indication of the agreement between observations and model predictions.

All five configurations presented good correlations with observations with an average $r$ of 0.752 although lower values were obtained for the five-layer schemes (0.695 ± 0.077 and 0.745 ± 0.044) than for the



configurations using the hybrid-layer scheme ($0.750 \pm 0.062$ - $0.801 \pm 0.034$). Among these multiple-layer-scheme simulations, the one showing the best $r$ is ECMWF_70 which was the configuration with the coarser spatial resolution for LOTOS-EUROS among the three. Therefore, regarding the degree of correlation for multilayer configurations, increasing horizontal resolution to very fine grid sizes in the photochemical model

5  does not improve results (providing that no changes are implemented in the emission inventory).

| | | r | | | | | FBIAS | | | | | NMSE | | | | |
|---|---|---|---|---|---|---|---|---|---|---|---|---|---|---|---|---|
| | | ECMWF_5 | ECMWF_70 | ECMWF_HR_70 | WRF_5 | WRF_70 | ECMWF_5 | ECMWF_70 | ECMWF_HR_70 | WRF_5 | WRF_70 | ECMWF_5 | ECMWF_70 | ECMWF_HR_70 | WRF_5 | WRF_70 |
| VILLA DEL PRADO | | 0.756 | 0.796 | 0.784 | 0.711 | 0.739 | 0.179 | 0.232 | 0.039 | 0.084 | 0.009 | 0.066 | 0.080 | 0.052 | 0.043 | 0.041 |
| SAN MARTÍN DE VALDEIGLESIAS | | 0.744 | 0.811 | 0.736 | 0.692 | 0.751 | 0.157 | 0.210 | 0.099 | 0.089 | 0.036 | 0.057 | 0.066 | 0.051 | 0.035 | 0.029 |
| EL ATAZAR | | 0.690 | 0.780 | 0.741 | 0.621 | 0.702 | 0.043 | 0.079 | 0.083 | 0.014 | 0.051 | 0.034 | 0.031 | 0.052 | 0.032 | 0.032 |
| SAN PABLO DE LOS MONTES | RURAL | 0.682 | 0.737 | 0.638 | 0.563 | 0.638 | 0.030 | 0.084 | 0.063 | 0.020 | 0.054 | 0.031 | 0.031 | 0.061 | 0.023 | 0.027 |
| CAMPISÁBALOS | | 0.683 | 0.747 | 0.496 | 0.618 | 0.684 | 0.313 | 0.358 | 0.168 | 0.264 | 0.211 | 0.140 | 0.162 | 0.142 | 0.116 | 0.088 |
| GUADALIX DE LA SIERRA | | 0.721 | 0.785 | 0.771 | 0.759 | 0.803 | 0.258 | 0.293 | 0.185 | 0.196 | 0.154 | 0.120 | 0.128 | 0.094 | 0.088 | 0.067 |
| ORUSCO DE TAJUÑA | | 0.750 | 0.822 | 0.785 | 0.690 | 0.747 | 0.029 | 0.120 | 0.025 | 0.005 | 0.071 | 0.045 | 0.040 | 0.040 | 0.027 | 0.034 |
| ALCORCÓN | | 0.748 | 0.807 | 0.801 | 0.736 | 0.793 | 0.176 | 0.282 | 0.035 | 0.042 | 0.020 | 0.110 | 0.127 | 0.130 | 0.075 | 0.075 |
| TOLEDO | | 0.702 | 0.738 | 0.709 | 0.735 | 0.771 | 0.139 | 0.224 | 0.072 | 0.103 | 0.017 | 0.081 | 0.097 | 0.081 | 0.059 | 0.047 |
| ENSANCHE DE VALLECAS | | 0.767 | 0.817 | 0.786 | 0.711 | 0.797 | 0.076 | 0.236 | 0.064 | 0.021 | 0.024 | 0.105 | 0.107 | 0.107 | 0.097 | 0.082 |
| VILLAVERDE | | 0.747 | 0.783 | 0.754 | 0.715 | 0.782 | 0.169 | 0.328 | 0.034 | 0.066 | 0.048 | 0.145 | 0.171 | 0.171 | 0.113 | 0.103 |
| ARTURO SORIA | | 0.735 | 0.815 | 0.763 | 0.605 | 0.778 | 0.231 | 0.388 | 0.083 | 0.066 | 0.094 | 0.181 | 0.207 | 0.180 | 0.151 | 0.118 |
| FAROLILLO | | 0.732 | 0.784 | 0.729 | 0.650 | 0.743 | 0.083 | 0.243 | 0.066 | 0.108 | 0.070 | 0.123 | 0.122 | 0.169 | 0.151 | 0.125 |
| PLAZA DEL CARMEN | URBAN | 0.609 | 0.727 | 0.714 | 0.476 | 0.704 | 0.373 | 0.526 | 0.228 | 0.183 | 0.223 | 0.358 | 0.389 | 0.283 | 0.287 | 0.230 |
| GUADALAJARA | | 0.710 | 0.808 | 0.764 | 0.757 | 0.812 | 0.227 | 0.313 | 0.199 | 0.175 | 0.107 | 0.125 | 0.141 | 0.101 | 0.084 | 0.060 |
| MOSTOLES | | 0.774 | 0.824 | 0.790 | 0.786 | 0.811 | 0.211 | 0.317 | 0.051 | 0.121 | 0.027 | 0.115 | 0.143 | 0.128 | 0.074 | 0.071 |
| ARANJUEZ | | 0.769 | 0.785 | 0.754 | 0.793 | 0.838 | 0.238 | 0.331 | 0.193 | 0.163 | 0.119 | 0.118 | 0.157 | 0.104 | 0.069 | 0.052 |
| RETIRO | | 0.737 | 0.822 | 0.778 | 0.598 | 0.788 | 0.184 | 0.342 | 0.036 | 0.010 | 0.031 | 0.156 | 0.171 | 0.156 | 0.163 | 0.109 |
| TRES OLIVOS | | 0.742 | 0.811 | 0.744 | 0.617 | 0.770 | 0.120 | 0.202 | 0.063 | 0.084 | 0.087 | 0.086 | 0.089 | 0.113 | 0.126 | 0.088 |
| AZUQUECA DE HENARES | | 0.771 | 0.838 | 0.804 | 0.773 | 0.813 | 0.182 | 0.269 | 0.111 | 0.126 | 0.051 | 0.092 | 0.109 | 0.072 | 0.069 | 0.053 |
| BARAJAS - PUEBLO | | 0.784 | 0.809 | 0.792 | 0.686 | 0.793 | 0.165 | 0.324 | 0.034 | 0.474 | 0.188 | 0.131 | 0.173 | 0.139 | 0.487 | 0.193 |
| RIVAS-VACIAMADRID | | 0.730 | 0.768 | 0.748 | 0.740 | 0.781 | 0.100 | 0.260 | 0.088 | 0.167 | 0.087 | 0.128 | 0.135 | 0.131 | 0.107 | 0.085 |
| JUAN CARLOS I | | 0.786 | 0.825 | 0.765 | 0.707 | 0.759 | 0.006 | 0.167 | 0.071 | 0.188 | 0.158 | 0.088 | 0.079 | 0.129 | 0.165 | 0.138 |
| EL PARDO | | 0.822 | 0.831 | 0.693 | 0.778 | 0.763 | 0.119 | 0.201 | 0.066 | 0.101 | 0.014 | 0.065 | 0.085 | 0.105 | 0.075 | 0.072 |
| ALGETE | | 0.807 | 0.864 | 0.825 | 0.718 | 0.787 | 0.084 | 0.166 | 0.038 | 0.031 | 0.050 | 0.047 | 0.051 | 0.053 | 0.042 | 0.042 |
| MAJADAHONDA | SUBURBAN | 0.782 | 0.832 | 0.717 | 0.717 | 0.734 | 0.066 | 0.174 | 0.019 | 0.027 | 0.051 | 0.065 | 0.065 | 0.104 | 0.064 | 0.077 |
| ILLESCAS | | 0.805 | 0.835 | 0.792 | 0.766 | 0.812 | 0.212 | 0.303 | 0.151 | 0.160 | 0.076 | 0.097 | 0.127 | 0.090 | 0.074 | 0.055 |
| TORREJON DE ARDOZ | | 0.784 | 0.839 | 0.820 | 0.761 | 0.811 | 0.134 | 0.293 | 0.132 | 0.103 | 0.033 | 0.119 | 0.137 | 0.102 | 0.090 | 0.077 |
| VALDEMORO | | 0.751 | 0.763 | 0.772 | 0.762 | 0.812 | 0.243 | 0.336 | 0.136 | 0.158 | 0.086 | 0.128 | 0.169 | 0.099 | 0.084 | 0.063 |
| CASA DE CAMPO | | 0.716 | 0.814 | 0.737 | 0.608 | 0.754 | 0.027 | 0.188 | 0.006 | 0.164 | 0.125 | 0.112 | 0.079 | 0.127 | 0.158 | 0.119 |
| *AVERAGE* | | 0.745 | 0.801 | 0.750 | 0.695 | 0.769 | 0.152 | 0.260 | 0.088 | 0.117 | 0.079 | 0.109 | 0.122 | 0.112 | 0.108 | 0.082 |
| *S. DEVIATION* | | 0.044 | 0.034 | 0.062 | 0.077 | 0.044 | 0.089 | 0.095 | 0.060 | 0.095 | 0.059 | 0.060 | 0.068 | 0.050 | 0.090 | 0.046 |

**Figure 2.** Values of the Pearson's correlation factor ($r$), fractional bias (FB) and normalised mean standard error (NMSE) of the comparison of the five LOTOS-EUROS simulations with observations of hourly $O_3$ data from 35 background monitoring stations located in the MAB. Colours illustrate the model agreement from blue (worst) to red (best).

LOTOS-EUROS tends to moderately overestimate ground-level $O_3$ concentrations varying widely in the five set-ups. It was clear that deviations with respect to observations declined with the use of a higher number of vertical levels and finer spatial resolution. The absolute values of the averaged FB for WRF_70 ($0.079 \pm 0.059$) and ECMWF_HR_70 ($0.088 \pm 0.060$) were substantially lower than those of the other

10  three configurations ($0.260 \pm 0.095 - 0.117 \pm 0.095$). Clear improvements were observed using finer spatial resolution in LOTOS-EUROS (either WRF simulations or ECMWF_HR_70) while ECMWF_5 and, espe-





cially, ECMWF_70 presented systematic but moderate overestimations (Figure S3). In consequence, the best configurations for adjusting the model bias were WRF_70 and ECMWF_HR_70.

A major reason for the overestimation detected for ECMWF runs with coarser spatial resolution was associated with an excessive $O_3$ formation in the noon hours of the day in situations of low wind speed as

shown in Figure 3. This plot shows the correlation between the model bias and the modelled wind speed in the location of El Retiro (see location in Table 2) in Madrid for the five runs. In the plots corresponding to ECMWF_5 and ECMWF_70 runs we observe systematic positive bias especially in the period 14–20 UTC when the formation is strong although it only spiked with low wind speed. This feature was not so marked in the three remaining configurations and, in particular, in the two WRF runs the bias values were randomly

distributed around zero. ECMWF_HR_70 run showed a subtler systematic overestimation during daytime but the correlation with low wind speeds was not observed in this case. Analysing the night-time period (0–6 UTC) we detect that the systematic overestimation was only present in the ECMWF_70 execution.

Regarding the simulation errors quantified via the NMSE (Figure 2), the results were satisfactory because the values of this parameter remained low for the five configurations (means ranging between $0.082 \pm 0.046$

and $0.122 \pm 0.068$ with WRF_70 run showing the best performance). Relevant information extracted from NMSE values in Figure 2 was that the errors of the model were consistently lower in rural areas ($0.063 \pm 0.038$) than in suburban ($0.096 \pm 0.033$) and urban ($0.135 \pm 0.077$) sites. This might be an effect of the simulation of the interaction between $O_3$ and $NO_x$, which acquires more relevance as a source of errors in the vicinity of traffic sources. Other authors (de la Paz et al., 2016) suggest that the use of an urban

canopy model (not used in this simulation) improves model predictions in densely built areas by reducing the overestimation of wind speed.

Systematic features characterised the modelled mean daily cycles of the 35 stations (Figure 4 shows the plots for the El Pardo site as a typical example). The daily cycles obtained with the simulations performed with the mixed-layer scheme of five levels (ECMWF_5 and WRF_5) presented a sharp increment in $O_3$

concentrations from 6 to 7 UTC which was not present in the simulations performed with the hybrid-scheme of 70 levels. In these runs, the morning increment of $O_3$ after the rush hour was delayed one or two hours (WRF_70 and ECMWF_HR_70 runs) and clearly smoothed. The observations reflected that the timing of the increase was better represented in the mixed-layer scheme runs although the increase was excessively abrupt. The occurrence of this steep increase in the concentrations in the executions performed with the





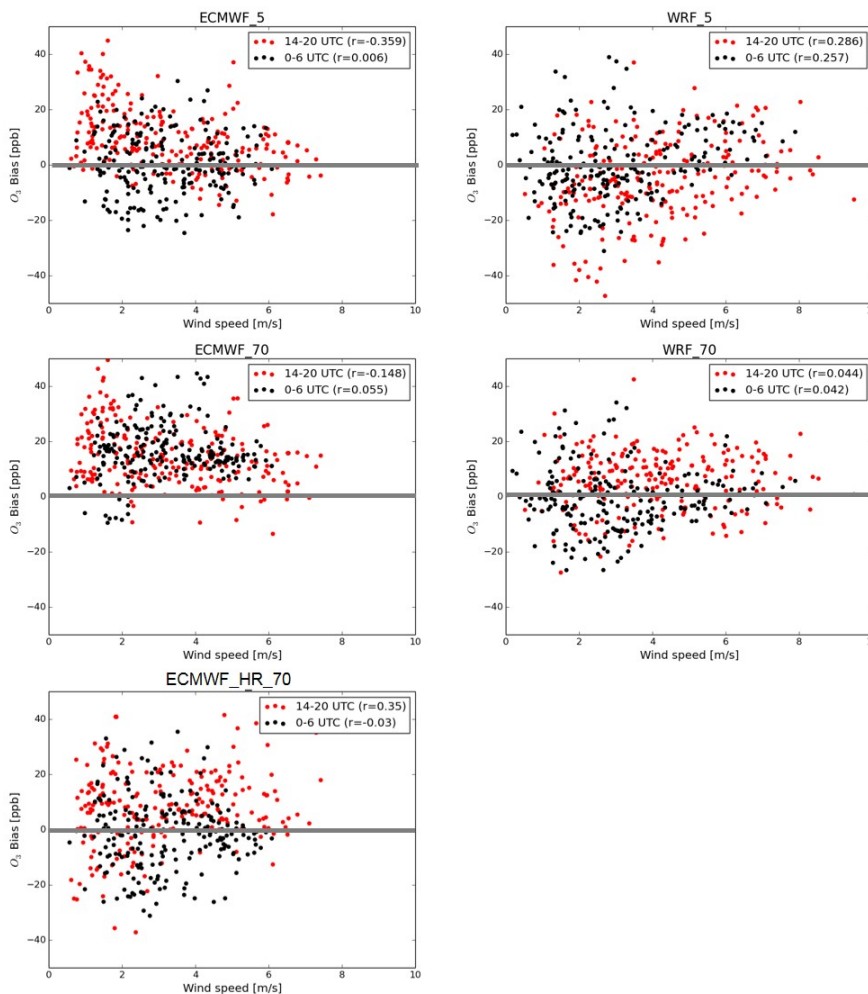

**Figure 3.** Correlation plots between the LOTOS-EUROS bias of surface $O_3$ and simulated wind speed at the station of Retiro disaggregated in two day periods: 0–6 and 14–20 UTC.

mixed-layer scheme coincided with the first steps of the boundary layer development. In the mixed-layer scheme, a rise of the boundary layer leads immediately to complete mixing of $NO_x$ emitted at the surface over the (increased) boundary layer, and thus limits titration of ozone; in the 70-layer schemes, however, the



mixing over the boundary layer seems to take place more gradually. A more extensive validation including other tracers than the chemically active ozone should provide insight into which scheme performs better under which conditions, and preferably lead to better characterization of the vertical diffusion.

### 3.1.2 O₃ vertical profiles

The evaluation of CTMs in the vertical direction has always been a difficult task due to the small number of high-resolution vertical observations to compare with. In this work, data from $O_3$ free soundings launched from Madrid airport at 12 UTC every seven days on 6, 13, 20 and 27 July 2016 were used for this purpose. Comparisons between the vertical profiles of modelled $O_3$ with the five different set-ups and the observations are presented in Figure 5. The corresponding profiles of wind direction and speed (modelled data are

taken from the input meteorological datasets) can be found in the supplementary information (Figure S4). As suggested by Querol et al. (2018), the enrichment of $O_3$ in the lower troposphere during episodes without ventilation is high as a consequence of the intense photochemical formation and the development of convective circulations. This typically results in vertical profiles in which $O_3$ concentrations are relatively high near the surface up to 2000 m a.s.l. at midday soundings in Madrid, as shown also here in Figures 5a and 5d.

During venting episodes, the more intense surface dispersion explains why $O_3$ vertical profiles have lower values in the mixing layer (Figures 5b and 5c).

  The event of 27 July was characteristic of an accumulation scenario with high concentrations near the surface while on 13 July (a typical venting event), $O_3$ in the lower levels was moderate and increased with altitude as described above. These two profiles were correctly simulated by most runs. However, the mod-

elled profiles for the 6 and 20 July showed overestimation in the lower levels with respect to observations and for the 20 July case, all the high-resolution simulations overestimated the observed values. In Figure S4, we can also check how the input meteorological data used to feed the simulations (ECMWF and WRF fields) closely reproduced the wind profiles obtained during the soundings of those four days both on speed and direction. Because of the complexity of the vertical mechanisms, further research should be conducted to

investigate the causes of this mismatch in some events.

  From the qualitative perspective, the first obvious conclusion was that a larger number of vertical levels in the model considerably improved the capability for capturing the vertical gradients of $O_3$ concentrations with the exception of the lowest level on 20 July. However, even in the simplest vertical scheme (five layers),





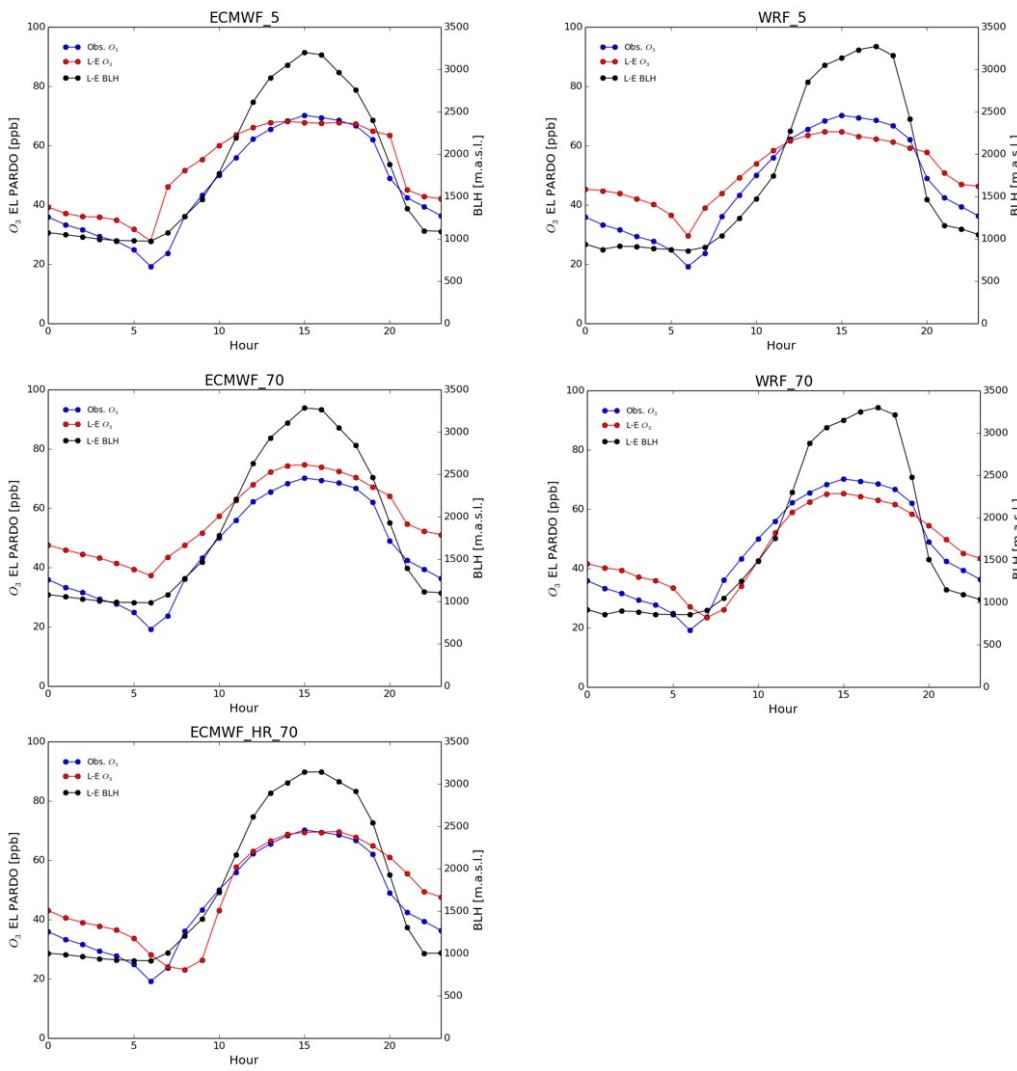

**Figure 4.** Simulated average diurnal cycles obtained from the five LOTOS-EUROS configurations compared with the mean cycle from the observations in the El Pardo background station. The modelled evolution of the boundary layer height is also shown.





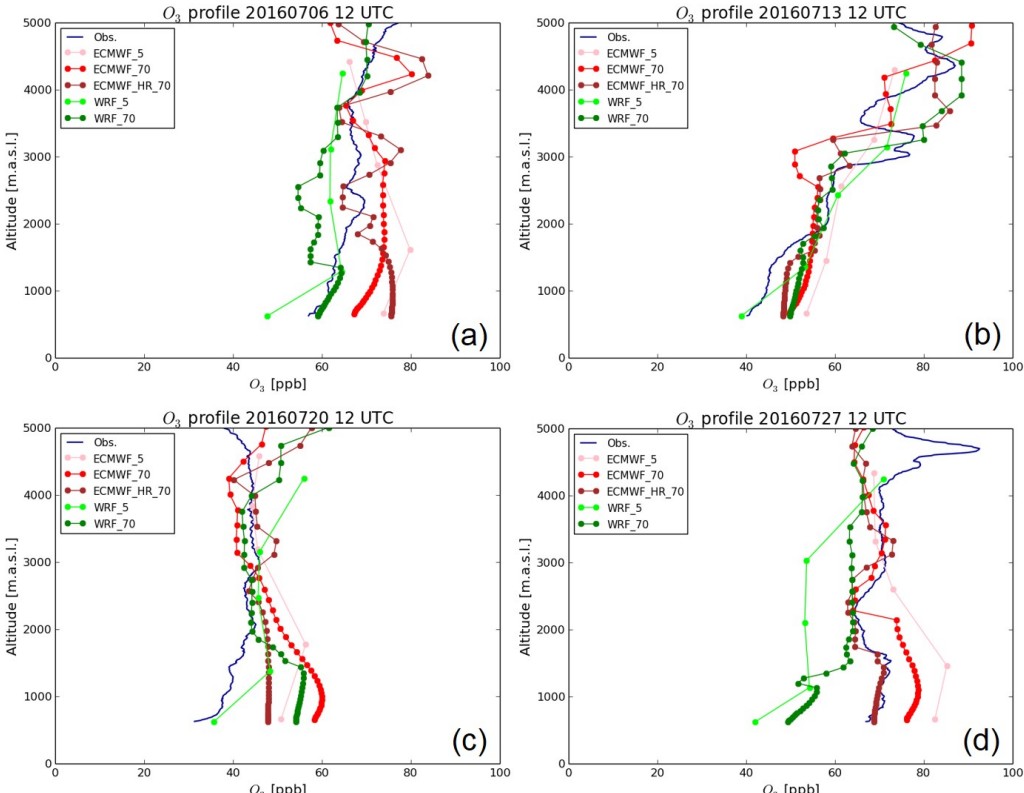

**Figure 5.** Real and simulated vertical profiles of $O_3$ for the 6, 13, 20 and 27 July 2016.

the model was able to reproduce the general vertical trends. A particular meteorological scenario was present
during July 13. We can observe two $O_3$ layers centred around 3000 and 4300 m a.s.l . The two multilayered
WRF runs captured these two layers at, approximately, the correct altitudes although the 3000 m layer was
not as marked as in the observations. The ECMWF runs also presented these two features but displaced in
5    altitude by around 200–300 m with respect to observations.

In all the WRF_5 runs and on 27 July WRF_70 simulations a steep drop of surface concentrations was
noticed. This is probably associated with the emission model configuration, the fine spatial resolution of the
runs (3 x 3 km$^2$) and the vertical mixing in these set-ups. The $O_3$ soundings were released from Adolfo





Suárez Barajas Airport, which is one of the major airports in Europe with more than 53 million passengers and 470 tonnes of goods transported in 2017 (http://www.aena.es/) and significant $NO_x$ emissions. The emission model employed by LOTOS-EUROS allocates at the surface level all the air traffic emissions in the grid cell where the airport is located. These emissions, added to those produced by road traffic, lead

to a grid cell that has the highest $NO_x$ emissions in the domain. As a consequence, and given the fine size of the grid cells in the WRF runs, $O_3$ levels were excessively reduced compared with reality in the aforementioned cases. Regarding ECMWF runs, the coarser spatial resolution did not allow observation of this feature because the modelled emissions from the airport were distributed on a larger surface.

   Quantitatively, the model generally showed the ability to reproduce the same order of magnitude of the

concentrations observed in the $O_3$ soundings at all altitude levels. The values of the statistical parameters that indicate the quality of the simulations (FB and NMSE) of the vertical profiles of $O_3$ are shown in Table 3 where generally satisfactory values can be observed with poorer results especially for July 20. The FB showed a majority of positive values indicating overestimation although, in most cases, it was moderate (the range of averages for the five configurations was –0.5 to 0.11). The NMSE data in Table 3 support this

conclusion because the average errors were small (0.023–0.042).

   Summarising, the configuration that presented the best overall performance among the five tested in the previous sections was WRF_70, so it was employed for interpreting the variability of $O_3$ in the MAB during July 2016.

## 3.2   Interpretation of $O_3$ in the MAB in July 2016

According to the dominant circulation over the MAB, three different episodes were distinguished and, with the aid of the model outputs, the basic features of the three events were described. Figure S2 shows the location of the selected monitoring stations used to characterise the behaviour of surface $O_3$ in the different sectors of the MAB.

### 3.2.1   Recirculation events (REC)

These events correspond to the pattern sketched by Plaza et al. (1997) in which wind direction turns clockwise during the day aided by the effect of the blocking effect of the Guadarrama range while Querol et al. (2018) described that the mixing layer growth at midday was reduced favouring vertical recirculation at the



eastern slopes of the Guadarrama range (see section 2.1). In July 2016 four REC periods were identified: 1–6, 8–11, 15–17 and 25–28. To illustrate the main features of REC episodes, the period 15–17 July will be used as an example (Figures 6 and 7). A complete pattern of simulated fields of $O_3$, wind and relative humidity (RH) for July 2016 can be consulted in Figure S5.

5     Surface wind speeds registered during REC episodes were weak (Figures 6 and S5) and the change in direction associated with recirculation is observed. However, despite the local circulation, air masses remain inside the basin during REC days aided by a relatively thin mixing layer at 12 UTC (Figure S4).

    A stable band of high RH centred at around 4000 m is observed in Figures 6 and S5 which can be associated with the evapotranspiration caused by the intense heating registered during these events. The 10 presence of a high-altitude trough located to the west of the IP during the 3–6 July REC period, induced moist south-westerlies at altitude resulting in the development of convective clouds in the evenings (Figure S5).

    Surface $O_3$ concentrations generally reach high values at the central time of the day during REC episodes (Figure 6). During the 17 REC days registered in July 2016, 37 exceedances of 180 $\mu g/m^3$ were recorded in all the monitoring stations in the study area. The sectors suffering the greatest impact of $O_3$ are the W–N belt 15 of the MMA (MAJ, PAR, COL, ALC and ALG) and more episodically, the Henares valley (AZU, AHE and GUA), the SE of the basin (ORT and VSA) or, when SW winds in the evening reached sufficient intensity (for example on 16 July), the $O_3$-enriched air masses reach rural stations located in the NE of the basin (ATA and CAM) late in the day. LOTOS-EUROS surface concentration maps show that at 12 UTC $O_3$ begins to rise, reaching the maximum concentrations around 18 UTC. Time series of $O_3$ also support this conclusion 20 because the highest $O_3$ levels are observed in the evening (Figure 6).

    To study the three-dimensional variability of $O_3$ during a typical REC event, Figure 7 presents longitudinal and latitudinal cross-sections of simulated $O_3$ and $NO_2$ for four different hours during 16 July. The use of $NO_2$ plots allows observation of the evolution of fresh emissions from the Madrid conurbation in the course of the day. This figure illustrates the strong photochemical formation of $O_3$ followed by the accumulation 25 during a typical REC episode. In the first hours of the day, a very shallow boundary layer combined with the stagnant conditions allows locally emitted precursors to accumulate inside the basin. Ozone is then effectively eliminated by titration with NO coming mainly from traffic emissions. This explains the steep drop in $O_3$ time series during the rush hours (Figure 6) with the exception of rural stations far from the Madrid conurbation like SPM, which showed a more stable behaviour in their $O_3$ concentrations.





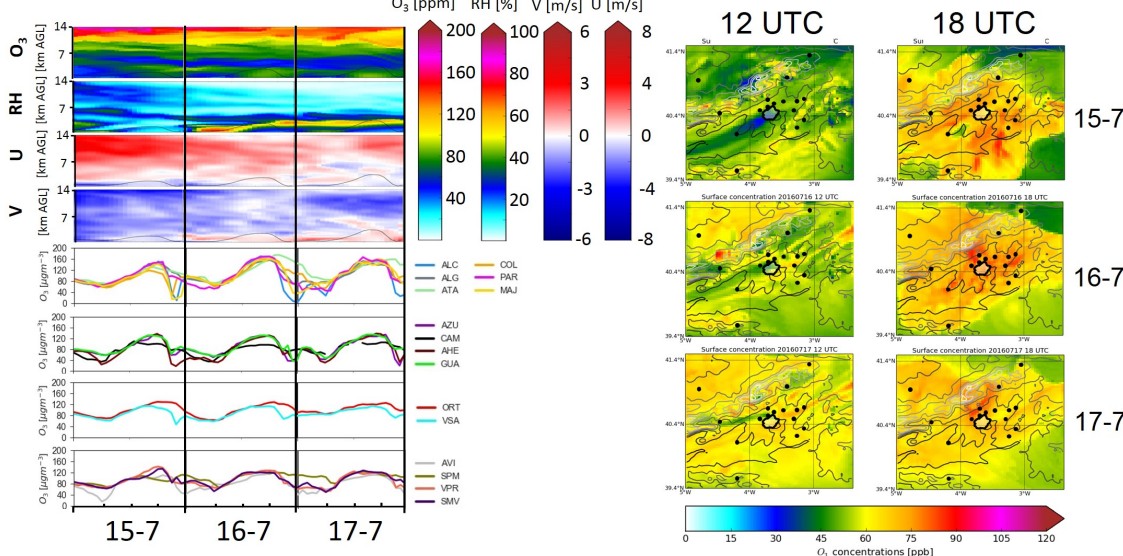

**Figure 6.** Left: Hourly $O_3$ concentrations recorded at selected monitoring stations in the MAB and simulated fields of $O_3$ concentration, relative humidity and U and V components of wind for the period 15–17 July 2016 over the centre of the MAB. Right: Surface $O_3$ concentration maps obtained from LOTOS-EUROS simulations at 12 and 18 UTC of 15, 16 and 17 July 2016.

At 18 UTC of 16 July, we can see how $O_3$ levels increased drastically, and the boundary layer depth grew up to 3500 m a.s.l. aided by convection at 18 UTC. Normally, REC events show higher planetary boundary layer (PBL ) heights in the evening. Figure 7 shows how the strong convection during REC events injected ground-level pollutants at high altitudes during the late afternoon and the evening reaching up to 3500 m a.s.l.

5   as illustrated in the $NO_2$ plots. When the night-time stable boundary layer forms after sunset, air masses with high $O_3$ that originated near the surface during the previous day were decoupled and remained in the residual layer at altitudes ranging between 2000 and 4000 m a.s.l. forming reservoir layers (00 and 06 UTC cross-sections in Figure 7) which can fumigate the following day. These reservoir layers can also be observed as a relatively thin band at an altitude of 2000–4000 m a.s.l. during every night of the REC period (Figures 6

10   and S5).





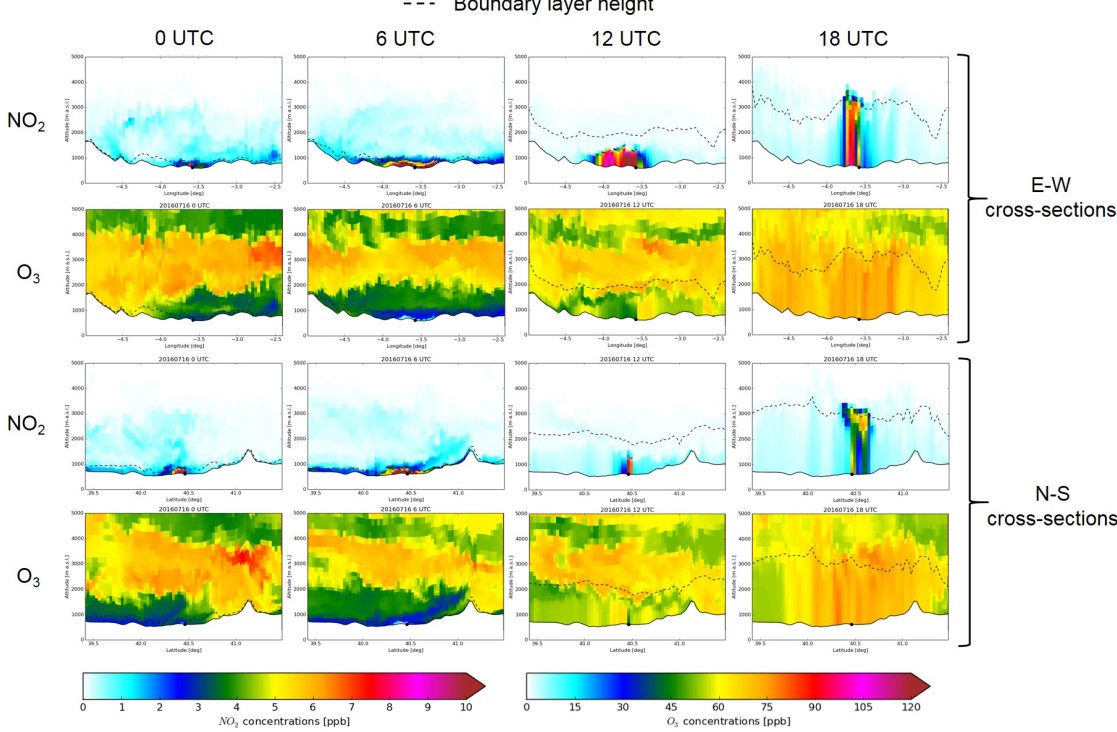

**Figure 7.** Longitudinal and latitudinal vertical cross-sections of $NO_2$ and $O_3$ for 16 July 2016.

### 3.2.2 Northern advective events (NAD)

We will refer to NAD events as those during which the dominant situation consisted of the advection of air masses coming from the north over the MAB. During July 2016, the following two periods matched that description: 12–14 and 22–24.

5    During NAD periods, surface wind is channelled following the NE–SW axis parallel to the Guadarrama range resulting in prevailing north-easterlies in the lowest tropospheric layers while in the upper levels the dominant component is NW (Figures 8, 9 and S5) often associated with the passage of cold fronts from the Atlantic. Winds are generally stronger than in REC events, which implies a renovation of air masses and lower temperatures.



Humidity during NAD events is conditioned by the arrival of air masses off the Atlantic, which are generally moist. During the period 12–14 July, a band of high RH (in the order of 50–60%) that reaches an altitude of approximately 3000 m a.s.l. can be observed (Figure 8). Moreover, a diagonal band of low RH can be observed descending from an altitude of 8000–9000 m a.s.l. at around 10 UTC on 12 July reaching

5 the surface by midday of 14 July. This structure was associated with layers of high $O_3$ detached from the lower stratosphere as shown in Figure 8. These low-RH stratospheric intrusions were observed also on the 22nd and 23rd of July during the third NAD episode (Figure S5).

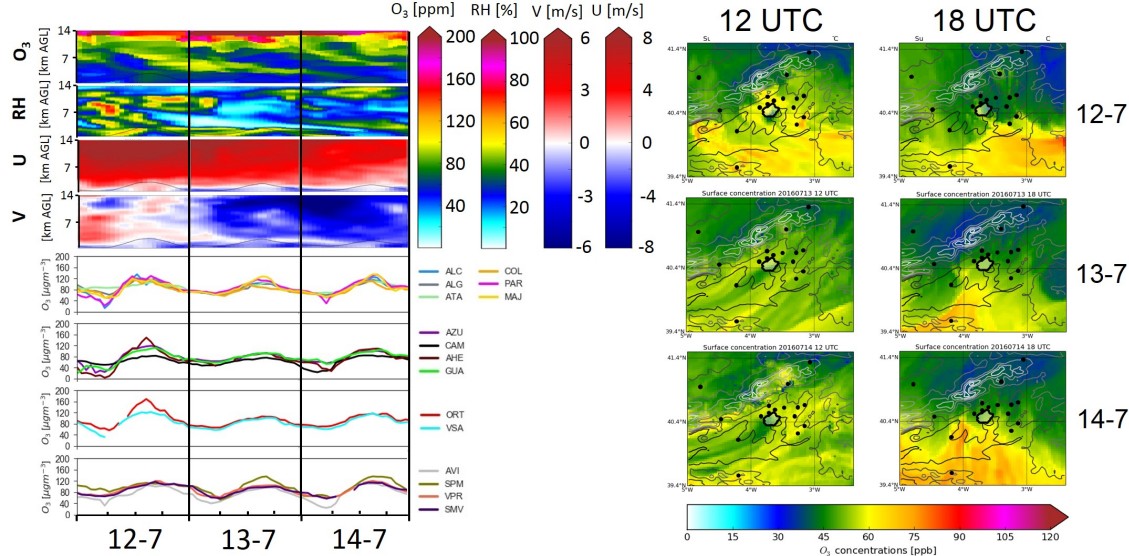

**Figure 8.** Left: Hourly $O_3$ concentrations recorded at selected monitoring stations in the MAB and simulated fields of $O_3$ concentration, relative humidity and U and V components of wind for the period 12–14 July 2016 over the centre of the MAB. Right: Surface $O_3$ concentration maps obtained from LOTOS-EUROS simulations at 12 and 18 UTC of 12, 13 and 14 July 2016 in the MAB.

The northern winds during NAD episodes push surface air masses with high $O_3$ towards the south and south-west of the MAB. However, ground-level $O_3$ concentrations are lower than those observed during REC

10 events. The highest $O_3$ levels were registered on 12 July at the south-eastern part of the domain (AHE and



ORT) with concentrations below 160 $\mu$g/m$^3$. Meanwhile, the maximum concentrations in the rest of stations that day fell below 130 $\mu$g/m$^3$, which is consistent with LOTOS-EUROS surface concentration maps for 12 July (Figure 8). The next two days, winds intensified and veered north resulting in concentrations that did not exceed 130 $\mu$g/m$^3$ along with noteworthy O$_3$ increments in the rural station of SPM located in the

southernmost part of the basin, in accordance with LOTOS-EUROS surface concentration maps of 13 and 14 July (Figure 8). Only one hourly exceedance of 180 $\mu$g/m$^3$ was registered during the six NAD days in all the air-quality stations in the study area in July 2016.

Figure 9 shows how increased advection reduced the residence time of polluted air masses over the region, resulting in lower and less variable O$_3$ concentrations throughout the day as observed in observations

presented in Figure 8. Moreover, the formation of reservoir layers during NAD episodes was less common due to the lower convection, relative to that of REC cases presented before.

It is also remarkable in Figure 9 that above 3000–3500 m a.s.l. O$_3$ concentrations were very high (in the order of 100 ppb according to the model). This was associated with the stratospheric intrusion of very dry air described above. LOTOS-EUROS reproduced this stratospheric intrusion that was detected from

data obtained with free and tethered O$_3$ soundings for the same period during a field campaign (Querol et al., 2018). The actual impact of this stratospheric intrusion on surface levels remains unclear because vertical cross-sections do not allow concluding if O$_3$ from this layer was effectively transported to the lowest tropospheric levels. In this particular case, the maximum altitude of the boundary layer according to LOTOS-EUROS reached its maximum values of 2500–2700 m a.s.l. (Figure 9) also limited by the wind ventilation

so, probably, the impact on the surface should be low (if any) in this case.

### 3.2.3   Southern advective events (SAD)

Southern transport implies the arrival of warm air masses, sometimes, coming from northern Africa (maximum temperatures at El Retiro during SAD events in the study period varied between 35.1 and 38.1 °C while during NAD periods they ranged from 27.1 to 35.1 °C). In July 2016, two SAD periods were observed: 18–

21 and 29–31. The first of these two periods has been chosen to illustrate the main features of SAD episodes (Figures 10–12).

SAD events are characterised by constant southerly winds at the surface and at altitude as shown in the simulated fields of U and V (Figures 10 and S5). Because the southern coast of the IP is a densely





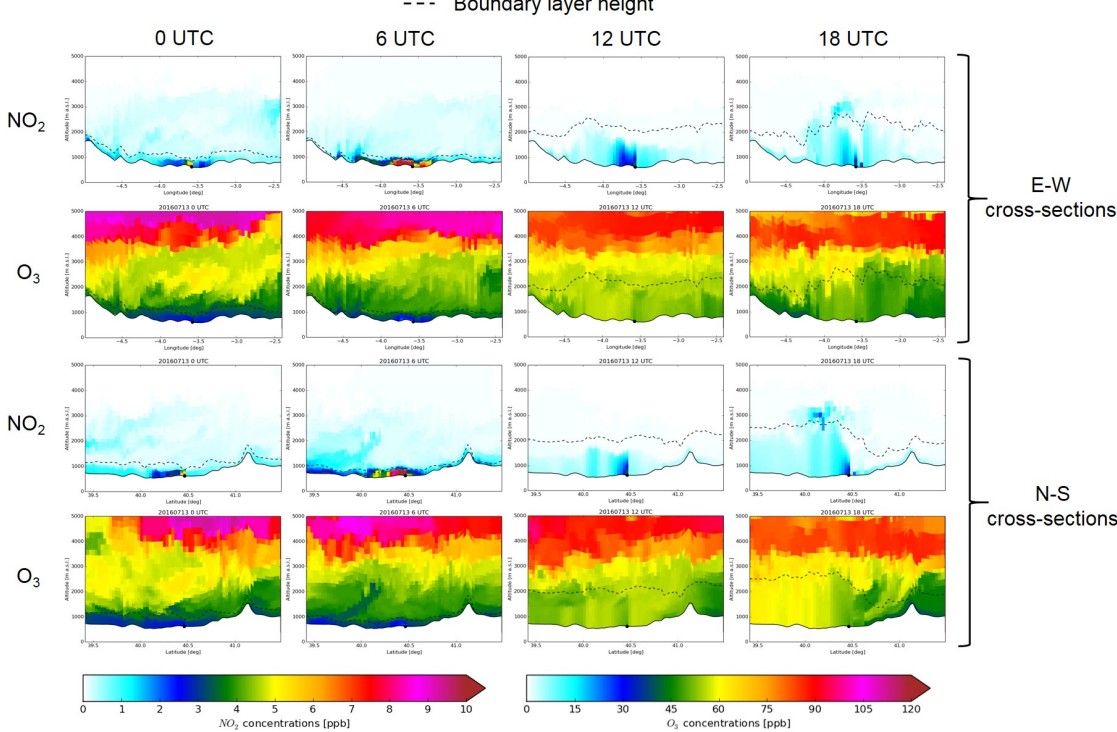

**Figure 9.** Longitudinal and latitudinal vertical cross-sections of $NO_2$ and $O_3$ for 13 July 2016.

populated area (especially in summer due to the strong touristic pressure) and anthropogenic emissions of $O_3$ precursors are high, including industrial emissions around the cities of Huelva, Seville and Algeciras, regional contribution of external $O_3$ may acquire importance at the basin during SAD events.

High RH values in the middle troposphere are observed during SAD periods (Figures 10 and S5) where
5   relevant increments were registered in the period 19–20 July. Although southerly winds are often associated with rain in the MAB, only small amounts of precipitation were collected during this period.

Analysing the 18–21 July case as a typical example of a SAD episode, we observe in the $O_3$ time series that concentrations increased in the entire basin earlier in the day than in the other two scenarios. While ground-level $O_3$ peaks around 18 UTC for REC conditions, daily maxima are found around 12 UTC under



SAD patterns, as confirmed in Figure 10. After the intense photochemical formation observed at midday, $O_3$ is transported towards the NE in the afternoon. This explains the concentration peaks registered at the rural stations located on that side of the basin (ATA and CAM) for example on July 19 or 29 (no figure shown of this day). SAD episodes are then periods in which $O_3$ produced at the MAB can be exported towards

5   the north of the IP. Less often, if winds blow from the SE in SAD events, $O_3$ can be transported across the Guadarrama range towards the NW as happened on 18 July (Figure 10).

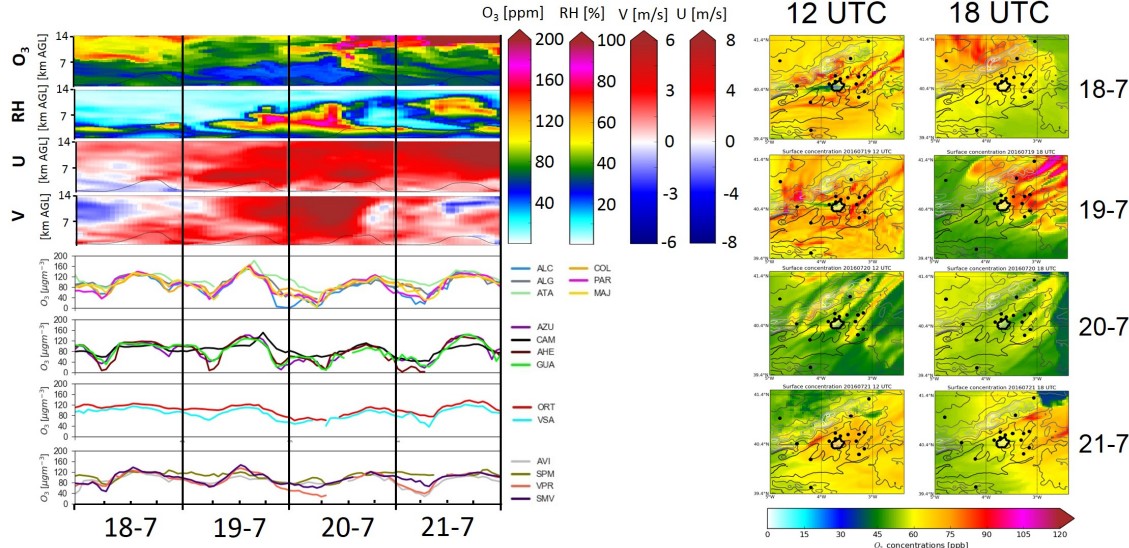

**Figure 10.** Left: Hourly $O_3$ concentrations recorded at selected monitoring stations in the MAB and simulated fields of $O_3$ concentration, relative humidity and U and V components of wind for the period 18–21 July 2016 over the centre of the MAB. Right: Surface $O_3$ concentration maps obtained from LOTOS-EUROS simulations at 12 and 18 UTC of 18, 19, 20 and 21 July 2016 in the MAB.

The effect of southern winds ventilating the area and some weak rainfall events limited the increase of $O_3$ although considerable levels were recorded in monitoring stations during SAD events. In general, $O_3$ concentrations during SAD periods are slightly lower than during REC events (with specific exceptions like

10   19 July) but higher than in NAD episodes. In the stations located in the basin hourly concentrations rarely



exceeded 180 $\mu g/m^3$ during SAD days (twice in July 2016 in the seven SAD days) but concentrations above 120 $\mu g/m^3$ were more frequent (1083 in total in all the stations in the study area or 155 per day) especially in the stations like ATA and CAM located on the NE of the basin (see 19 July in Figure 10). This proportion of records above 120 $\mu g/m^3$ is higher than during NAD events (average of 56 per day ) and lower than the rate registered during REC events (203 per day ).

Vertical cross-sections of $O_3$ and $NO_2$ on two consecutive days (18 and 19 July) from a SAD period have been used to illustrate the different behaviours observed (Figures 11 and 12). The intense accumulation of $NO_2$ observed in the 00 and 06 UTC plots points out that on these days, ventilation was not as effective as in NAD events. As a consequence, the $O_3$ daily cycles showed a considerable drop associated with titration in the morning rush hour unlike on NAD days and closer to the situation of REC episodes (Figure 10). Likewise, for NAD events, vertical mixing is limited as shown in the $NO_2$ vertical cross-sections of 18 and 19 July preventing the formation of reservoir layers during SAD events. The higher $O_3$ registered on 19 July seems to be related to the fact that a deeper boundary layer (maximum heights above 4000 m a.s.l. on 18 July, 3200 m a.s.l. on 19 July) allowed larger dilution, lowering surface concentrations.

### 3.2.4 The role of the Boundary Layer Height (BLH)

Figure 13a presents a comparison between observed (data from Querol et al. (2018)) and simulated (WRF_70 configuration) BLH at 12 UTC. We can observe that the model tends to overestimate the BLH at midday although the general trends are captured. In particular, the gradual decrease in the 12 UTC BLH from 11 to 14 July allowed $O_3$ to accumulate smoothly in the basin, which was described in the aforementioned work, is also observed in the simulated data. The overestimation is slight on most days although larger differences are observed in certain periods (5–10 and 29–31 July).

Querol et al. (2018) describe lower midday BLH in $O_3$ accumulation episodes (equivalent to REC events described here) than in venting episodes (NAD or SAD). The simulations with LOTOS-EUROS confirm that finding as observed in Figure 13b. The mixing layer is deeper on 13 July (NAD) than on 16 July (REC) from 0 to 16 UTC which includes the period of most effective photochemical formation of $O_3$. This allows a more effective formation of reservoir layers during REC events that fumigate to the surface as the diurnal convective circulation develops. After 16 UTC the BLH during the REC event grows higher than during the NAD day due to the larger convection in the second scenario.



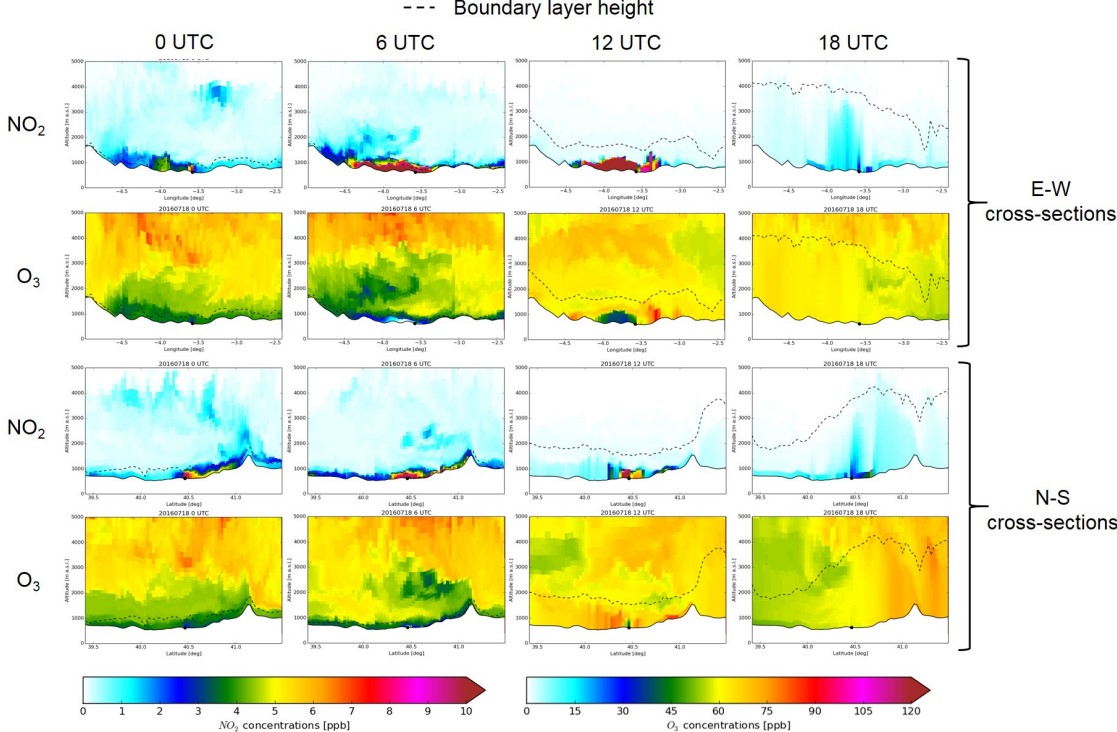

**Figure 11.** Longitudinal and latitudinal vertical cross-sections of NO$_2$ and O$_3$ for 18 July 2016.

## 4  Conclusions

Evaluation of a CTM is a basic tool for the analysis and forecasting of photochemical processes that give rise to high concentrations of tropospheric O$_3$ that frequently occur in the Mediterranean in summer. A preliminary requirement for the application of CTM for policy decisions is that they could reproduce adequately the processes and mechanisms identified by the field campaigns and reasonably reproduce the observations of the monitoring stations, especially during acute O$_3$ episodes.

In this work, we present the results obtained from a simulation exercise (July 2016) performed with the LOTOS-EUROS CTM over the MAB, representative of summer conditions. Five configurations with different combinations of spatial resolution (25 x 25 and 3 x 3 km$^2$), input meteorological data (ECMWF 7





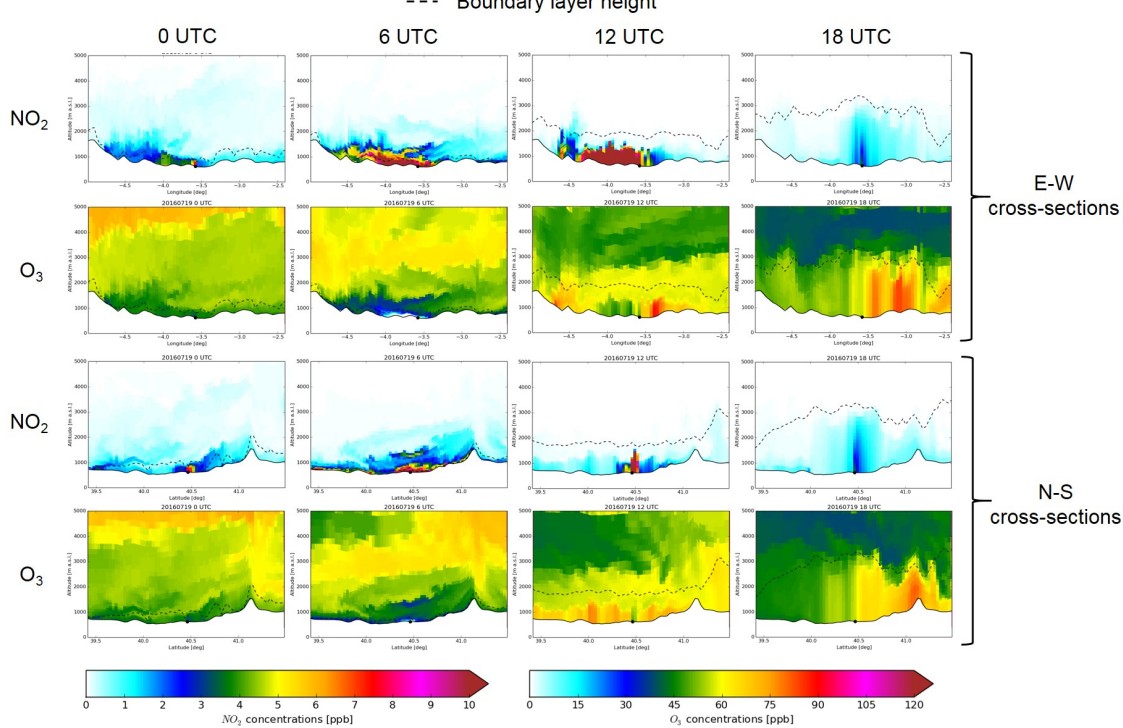

**Figure 12.** Longitudinal and latitudinal vertical cross-sections of $NO_2$ and $O_3$ for 19 July 2016.

x 7 km$^2$ for the IP and WRF, 1 x 1 km$^2$ for the MAB) and vertical structures (mixed-layer scheme with five altitude levels and hybrid-layer scheme with 70 altitude levels) for model evaluation and optimisation.

Our results show that the LOTOS-EUROS model performs in a satisfactory manner in the five set-ups. However, regarding surface $O_3$, it is clear that the model benefits from finer spatial resolutions in the hori-
5 zontal and also from the use of multilayered vertical schemes. As a result, WRF_70 and ECMWF_HR_70 were the optimal configurations.

Using multilayered 70 level set-ups, LOTOS-EUROS was able to reproduce the vertical gradients of $O_3$ in the Madrid basin, although in some cases the model presented an overestimation in the lower levels with respect to observations. In most cases, the model was also able to reproduce features like fine $O_3$ layers.



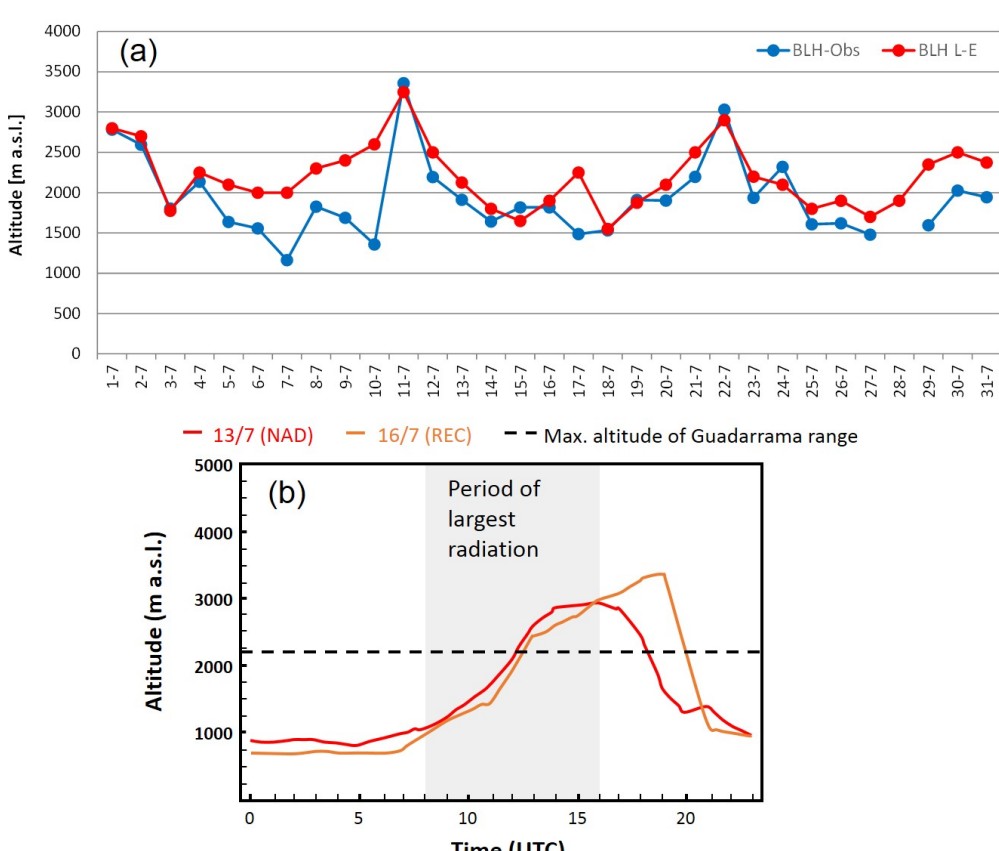

**Figure 13.** (a) Time series of the estimated and modelled midday BLH over Madrid airport for July 2016. Estimations calculated from the daily AEMET radio-soundings using the simple parcel method. (b) Modelled BLH for the 13 (NAD event) and the 16 (REC event) July 2016.

The performance of LOTOS-EUROS was partly successful, differentiating the vertical structure of $O_3$ under distinct meteorological conditions so further research is needed to improve CTMs' performances in this particular aspect with, for example, comparisons with data from $O_3$ soundings under different meteorological scenarios.



Therefore, the modelling system is suitable to be employed for the interpretation of $O_3$ variability in the region. In light of the present study, we suggest using vertical schemes of CTMs with a sufficient number of levels for capturing $O_3$ variability in the simulations of summer episodes in the Mediterranean region.

Employing the WRF_70 configuration of LOTOS-EUROS which has shown the best performance simu-
5 lating surface and vertical concentrations of $O_3$ in the MAB, we interpreted the variability of $O_3$ in the region. Three episode types have been identified regarding the dominating circulation. Two of them are associated with advection, either from the north (NAD) or from the south (SAD), while the third is associated with lo-cal/regional recirculation of air masses (REC). REC events are characterised by low winds that veer during the day from NE to SW following the axis of the Guadarrama range. These stagnant conditions combined
with the strong insolation and temperature registered during REC events favour the strong photochemical production and accumulation of $O_3$ over the MAB, likely to exceed 180 $\mu g/m^3$ during the afternoon and evening . Moreover, the strong convection helps to form reservoir layers located at 2000–4000 m a.s.l. which contribute to increasing surface $O_3$ in the following days when the upper limit of the boundary layer reaches those altitudes.

Marked differences have been found between the two venting episodes. During SAD episodes, winds are weak and external $O_3$ contributions from the south and south-east of the IP can be relevant, while in the case of NAD events, winds were generally stronger, favouring ventilation. As a consequence, surface $O_3$ during NAD events did not grow excessively (except in specific cases when wind speed is low). During SAD conditions, higher base $O_3$ concentrations (>120 $\mu g/m^3$) were registered but the 180 $\mu g/m^3$ threshold is
exceeded rarely in both episode types. One of the factors for this is the existence of a steady wind direction avoiding an effective accumulation of $O_3$ in reservoir layers. Both NAD and SAD events are associated with $O_3$ exportation to other air basins on the IP like the Ebro valley (to the NE) and the Tagus and Guadiana valleys (to the SW).

Intrusions of stratospheric $O_3$ have been observed with LOTOS-EUROS simulations in the form of bands
with a high concentration of $O_3$ and very low humidity. It is unclear whether these intrusions have an impact at the surface and, if so, what is the exact contribution to the $O_3$ observed there. Specific model-based analyses of these episodes should be performed to evaluate their actual impact on surface $O_3$ in the MAB.

The results from this study can be useful to understand the phenomenology of high $O_3$ episodes in the MAB and to gain knowledge to design appropriate strategies for air-quality management. Further research





must be implemented to investigate aspects like the sensitivity to emission reduction scenarios or the role of VOCs with emphasis on the biogenic ones. Moreover, to perform the tasks of validating and optimising CTMs, increasing efforts should be made to conduct more field campaigns in different air basins in the Mediterranean using state-of-the-art equipment to generate data and knowledge about $O_3$ behaviour both on the surface and vertically. Useful parameters to be included in these campaigns are $O_3$, $NO_x$, VOC and, when possible, intermediate products like $NO_y$, $HNO_3$ and $H_2O_2$ that, according to previous experience (Sillman, 1995), are key parameters for facing model-based $NO_x$–VOC sensitivity studies and the assessment of emission inventories.

In future, similar simulations to the one presented in this study should be performed in the different air basins in the IP where $O_3$ exceedances have been recorded (Querol et al., 2016). CTMs should be configured specifically for each region or air basin to assure the best performance by capturing the influence of topography and local circulations. For such studies, we highlight the importance of conducting experimental campaigns that can support the necessary model evaluation.

Finally, it should be noted that when running such fine resolutions for real applications it is also important to work on the emission datasets (out of the scope of this work). Increasing the detail in emission inventory (mainly based on a bottom-up approach) could improve the performance of CTMs when assessing sensitivities or emission scenarios.

*Author contributions.* Dr. Escudero conceived the presented idea, carried out the simulations with LOTOS-EUROS, collected and treated experimental data used for model evaluation and wrote the manuscript. Drs. Segers, Kranenburg and Schaap supervised LOTOS-EUROS simulations, modified the model code for performing runs with the different configurations and contributed in the post-processing of model outputs. Drs. Borge and de la Paz performed and validated simulations with WRF. Drs. Querol, Alastuey and Gangoiti contributed especially in the interpretation of $O_3$ phenomenology. All authors discussed the results and contributed to the final manuscript.

*Competing interests.* The authors declare that they have no conflict of interest.



*Acknowledgements.* This work was funded by the Ministry of Economy, Industry and Competitiveness and FEDER funds through the project HOUSE (CGL2016-78594-R), the Ministry of Agriculture, Fishing, Food and Environment, the Madrid City Council, the Madrid Regional Government and by the Department of Research, Innovation and University of the Aragón Regional Government and the European Social Fund (project E23_17D). The study was also partially supported by the scientific programme TECNAIRE-CM funded by the Directorate General for Universities and Research of the Greater Madrid Region (S2013/MAE-2972). The authors gratefully acknowledge air-quality data provision by the following entities: MITECO, Madrid City Council, AEMET and the Autonomous Communities of Madrid, Castilla León and Castilla La Mancha. Dr Escudero received a grant from the José Castillejo programme of the Ministry of Education and Science of Spain (ref. CAS17/00108) for a 6-month research visit at TNO.





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



| Name | Network | Type | Lat. (°) | Long. (°) | Alt. (m a.s.l.) |
|---|---|---|---|---|---|
| VILLA DEL PRADO | MAC | RURAL | 40.25 | –4.27 | 469 |
| SAN MART'IN DE VALDEIGLESIAS | MAC | RURAL | 40.37 | –4.40 | 707 |
| EL ATAZAR | MAC | RURAL | 40.91 | –3.47 | 995 |
| SAN PABLO DE LOS MONTES | EMEP | RURAL | 39.55 | –4.35 | 917 |
| CAMPISÁBALOS | EMEP | RURAL | 41.27 | –3.14 | 1360 |
| GUADALIX DE LA SIERRA | MAC | RURAL | 40.78 | –3.70 | 852 |
| ORUSCO DE TAJUÑA | MAC | RURAL | 40.29 | –3.22 | 795 |
| ALCORCÓN | MAC | URBAN | 40.34 | –3.83 | 693 |
| TOLEDO | CLM | URBAN | 39.87 | –4.02 | 500 |
| ENSANCHE DE VALLECAS | MCO | URBAN | 40.37 | –3.61 | 630 |
| VILLAVERDE | MCO | URBAN | 40.35 | –3.71 | 593 |
| ARTURO SORIA | MCO | URBAN | 40.44 | –3.64 | 698 |
| FAROLILLO | MCO | URBAN | 40.39 | –3.73 | 625 |
| PLAZA DEL CARMEN | MCO | URBAN | 40.42 | –3.70 | 657 |
| GUADALAJARA | CLM | URBAN | 40.63 | –3.17 | 620 |
| MÓSTOLES | MAC | URBAN | 40.32 | –3.88 | 650 |
| ARANJUEZ | MAC | URBAN | 40.04 | –3.59 | 512 |
| RETIRO | MCO | URBAN | 40.41 | –3.69 | 672 |
| TRES OLIVOS | MCO | URBAN | 40.50 | –3.69 | 715 |
| AZUQUECA DE HENARES | CLM | URBAN | 40.57 | –3.26 | 600 |
| BARAJAS-PUEBLO | MCO | URBAN | 40.47 | –3.58 | 631 |
| RIVAS-VACIAMADRID | MAC | SUBURBAN | 40.36 | –3.54 | 610 |
| JUAN CARLOS I | MCO | SUBURBAN | 40.47 | –3.61 | 669 |
| EL PARDO | MCO | SUBURBAN | 40.52 | –3.77 | 700 |
| ALGETE | MAC | SUBURBAN | 40.59 | –3.50 | 721 |
| MAJADAHONDA | MAC | SUBURBAN | 40.45 | –3.87 | 722 |
| ILLESCAS | CLM | SUBURBAN | 40.12 | –3.83 | 548 |
| TORREJÓN DE ARDOZ | MAC | SUBURBAN | 40.46 | –3.48 | 581 |
| VALDEMORO | MAC | SUBURBAN | 40.19 | –3.68 | 610 |
| CASA DE CAMPO | MCO | SUBURBAN | 40.42 | –3.75 | 645 |
| ÁVILA | CL | SUBURBAN | 40.66 | –4.70 | 1150 |
| ALCOBENDAS | MAC | URBAN | 40.54 | –3.64 | 671 |
| COLMENAR VIEJO | MAC | URBAN | 40.67 | –3.77 | 905 |
| ALCALÁ DE HENARES | MAC | URBAN | 40.48 | –3.38 | 589 |
| VILLAREJO DE SALVANÉS | MAC | URBAN | 40.17 | –3.28 | 761 |

**Table 2.** Details of the air-quality monitoring stations selected for this study. With the exception of the last five, which are classified as traffic sites, all the stations included in this table are located in background locations. Network codes: MCO, Madrid Council; CL, Castilla y León region; CLM, Castilla La Mancha region; MAC, Madrid region and EMEP.





|  | ECMWF_5 | ECMWF_70 | ECMWF_HR_70 | WRF_5 | WRF_70 |
|---|---|---|---|---|---|
| *Fractional Bias (FB)* | | | | | |
| **06-July-2016** | 0.10 | 0.10 | 0.13 | −0.08 | −0.04 |
| **13-July-2016** | 0.05 | 0.05 | 0.05 | 0.01 | 0.08 |
| **20-July-2016** | 0.17 | 0.25 | 0.16 | 0.13 | 0.22 |
| **27-July-2016** | 0.08 | 0.05 | −0.03 | −0.24 | −0.17 |
| *Normalised mean square error (NMSE)* | | | | | |
| **06-July-2016** | 0.024 | 0.015 | 0.030 | 0.009 | 0.007 |
| **13-July-2016** | 0.015 | 0.024 | 0.015 | 0.002 | 0.016 |
| **20-July-2016** | 0.057 | 0.116 | 0.040 | 0.026 | 0.086 |
| **27-July-2016** | 0.021 | 0.011 | 0.005 | 0.071 | 0.039 |

**Table 3.** FB and NMSE of the comparisons between LOTOS-EUROS simulations of vertical profiles of $O_3$ and data from four $O_3$ soundings performed in Madrid in July 2016.