# Peer review of "ANALYSIS OF SUMMER O3 IN THE MADRID AIR BASIN WITH THE LOTOS-EUROS CHEMICAL TRANSPORT MODEL"

_Atmospheric Chemistry and Physics, 2019_

## Referee Comment (RC1) · Anonymous Referee #2 · 18 Jul 2019

The article presents a detailed case study of how a state-of-the-art chemistry-transport model (Lotos-Euros) captures ozone air pollution episode, making use of an intense field campaign previously published in ACP. The presentation quality of the article is outstanding and the scientific discussions, both in terms of atmospheric processes and modelling challenges, is of very high level. Therefore, I recommend publication after the following points are addressed.

General Comments:

Over the past few years, the majority of model evaluation papers have been focused on long term simulations (at least annual), for statistical robustness considerations. But

this tendency comes at a cost: that few authors analyzed in details air pollution episode case studies. In that context, the article from Escudero et al. is a welcome initiative.

My only major comment would be that I somehow disagree with the authors that model evaluation is only useful to build confidence in the tools: it is also essential to guide their development. The detailed analysis presented here could thus be more conclusive in pointing specific issues that would deserve higher priority in future model development, i.e. being more specific that pointing to "future research" (as stated p26). Taking the example of vertical resolution, the two configurations tested here are somewhat extreme: going from 5 to 70 layers, which is presumably not realistic in long term simulations used in policy support or air quality forecasting. It would have been useful to know to what extent the 5 layer model captures the episode processes discussed in 3.2, and how a tradeoff could be found.

Specific comments:

P2L12 : why using plural here? only O3 is a secondary air pollutant

P5L14: the air pollution regimes (REC/SAD/NAD) should be introduced here and related to synoptic meteorological situations.

P6L10: indicate the range explored in terms of O3 dry deposition velocities. Was this impact assessed on the basis of free-tropospheric total ozone burden as in Stevenson et al. or rather surface ozone?

P7L2: what is the reference year for emissions used?

P9L5: the worsening of correlations when increasing resolution has been documented as "double penalty" in the field of meteorological forecast. It would be worth discussing in more detail the issue here

P12L2: it is indeed frustrating that NO2 is not included in the detailed validation. It would also be interesting to see some basing meteorological validation, not clear why WRF outperforms IFS and there could be compensation of errors

P16L25: how do you explain the local minima of O3 around 4km agl?

P18, Fig 7: what is driving the sharp horizontal convergence of NO2 between 12 and 18UT?

P20L15-20: this section deserves further discussion in light of Querol et al. 2018 (Section 4), where they analyze the likelihood of impact at the ground of the STE event in relation with diurnal PBL variability.

P21L9: The model still produces a peak in the mid/late afternoon and the shift of the O3 daily maxima is biased by the spatial distribution of stations, which miss the plume that has been advected NW at 18UT.

Figure 6, 8, 10: those figures are very nice and comprehensive. The only missing information is modelled O3 time series. Although model and observations are compared in the in 3.1.1 and 3.1.2, I am missing a visual comparison of time series. The vertical cross sections are difficult to read and would benefit from being consistent with Fig 7&9 (i.e. both could extend to 7km to allow discussing the stratospheric intrusion). The color legend of O3 for the vertical cross section should be consistent with the maps (and the label corrected: it is probably ppb, not ppm).

---

## Referee Comment (RC2) · Anonymous Referee #3 · 4 Aug 2019

The article presented here proposes an evaluation of the LOTOS-EUROS model through the implementation of 5 different configurations of the model, for the simulation of the ozone concentration fields, in the Madrid region. The 5 configurations are discriminated by the use of different horizontal and vertical resolutions, either for meteorological calculations (ECMWF or WRF), or for the implementation of the LOTOS-EUROS simulation itself. The document is very well written, the subject well exposed and the analysis of the episodes and their simulation is comprehensive. But we regret that - in the introduction - the description of the scientific issue is somehow reduced to the evaluation of a model over a region. It seems important to reposition the problem within a broader framework, which is that of understanding the phenomena of produc-

tion and transport of ozone. There have been many modeling studies on the subject, and in particular on the Mediterranean. And here, we miss the following information (which could help to better estimate the scientific impact of the paper) : what is the state of the art terms of fine restitution of the fields of ozone in the Mediterranean? what has been undertaken at this scale so far? What are the current shortcomings in terms of model performance for ozone, what are the locks for the restitution of fine-scale ozone fields, on the horizontal and on the vertical ? What is required to go further in terms of ozone modelling ? In what way does this fine-scale study bring new elements, not only to the evaluation of the model itself, but also to research on the subject? Also, why do the authors only focus on model resolution?

In the conclusions, there is a lack of discussion about the importance of improvements (improved indicators), with regard to the requirements of enhanced configurations (especially CPU time). It would also be important to give a strategy to choose a future configuration: is it reasonable to choose so many vertical levels (70)? Also, still for a contextualization (this time of the results and not of the stakes), the interpretations of the episodes (which are precise), lack some context. There has been indeed a lot of studies of ozone formation episodes in the literature. The formation of ozone downwind sources, and its dependence upon wind speed and vertical dilution are known: what exactly are the new knowledge at the end of the study, concerning the phenomenology of episodes and their properties, or concerning the ability of a CTM to simulate them? How can these new knowledge improve air quality management strategies? Are we just talking about improved scores from one version to another, or do some configurations allow to depict specific features that do not appear in the others ?

Specific comments :

Page 10 – line 5 - "In the plots corresponding to ECMWF_5 and ECMWF_70 runs we observe systematic positive bias especially in the period 14–20 UTC when the formation is strong although it only spiked with low wind speed. This feature was not so marked in the three remaining configurations and, in particular, in the two WRF runs

the bias values were randomly distributed around zero." How do the authors explain the mid-day biases of ECMWF compared to WRF? Is it just a problem of resolution of the meteorological calculations, or does it depend on the meteorological model itself?

Line 15 page 15: the best overall performance is analyzed on which criteria? Which parameters are used to affirm that the simulation is better? Should it be the restitution of the diurnal peak, the phasing of the morning increase, the total amount on the vertical, or just the indicator average...? In particular, the WRF70 has a strong underestimation of diurnal ozone in Figure 4 at El Pardo and this is still considered as the "best run". What about this feature at other stations?

The difference between what is observed and simulated is not always specified, even if we can guess it. Example: in "Figure 7. Longitudinal and latitudinal vertical cross-sections of NO2 and O3 for 16 July 2016". Same for figure 9 and figure 11. Also, for page 17 lines 1 to 10, it is important to specify that it is a vision drawn by the model and not the result of observations.

Figures 6, 8, 10: The location / typology of the station groups should be mentioned.

Figure 7: it would be more convenient for the reader to visualize on a map the latitudinal and longitudinal cuts.

[Figure]

---

## Author Comment (AC1) · 1 Oct 2019

*Response to reviewer #2 "Analysis of summer O3 in the Madrid air basin with the LOTOS-EUROS chemical transport model" by Miguel Escudero et al.*

**The authors would like to acknowledge the comments from reviewer #2. We thank the positive comments. All the suggestions demonstrate a good knowledge of the scientific field and that has resulted in a great improvement of the paper after his/her revision.**

*My only major comment would be that I somehow disagree with the authors that model evaluation is only useful to build confidence in the tools: it is also essential to guide their development. The detailed analysis presented here could thus be more conclusive in pointing specific issues that would deserve higher priority in future model development, i.e. being more specific that pointing to "future research" (as stated p26). Taking the example of vertical resolution, the two configurations tested here are somewhat extreme: going from 5 to 70 layers, which is presumably not realistic in long term simulations used in policy support or air quality forecasting. It would have been useful to know to what extent the 5 layer model captures the episode processes discussed in 3.2, and how a tradeoff could be found.*

**This is an interesting comment by the reviewer. The authors fully agree with the idea of the results evaluation should serve to trigger the improvements in air quality modelling. In this work, however, we aimed to analyse specific summer episodes of O3 in the MAB with the aid of high resolution simulations performed with LOTOS-EUROS. This means, that the main objective of the paper is to provide a phenomenological interpretation of O3 events in the area. It is also true that we make a detailed evaluation of the best configuration of the model for the specific area and period and, for that, we used, along with standard surface observations, highly resolved vertical profiles for O3.**

**Regarding the specific question of the reasonable number of vertical levels in the model configuration, the answer is that it is dependent on the objective of the study. In this study the environmental analysis was the main objective and it was logical and feasible from the perspective of CPU time to employ a considerable number of vertical levels because it allowed a better representation of the vertical variability of O3. In other studies in which CPU time is limiting such as air quality forecasting or long term analyses, the reasonable number of levels can be less.**

**This has been indicated in the conclusions:**

**"The main objective of the paper is to provide a phenomenological interpretation of O3 events in the area after performing a detailed evaluation of the best configuration of the model for the specific area and period. Regarding the specific question of the reasonable number of vertical levels in the model configuration, it is dependent on the objective of the study. In this study the environmental analysis was the main objective and it was logical and feasible from the perspective of CPU time to employ a considerable number of vertical levels because it allowed a better representation of the vertical variability of O3. In other studies such as air quality forecasting or long term analyses in which CPU time may be large, the reasonable number of levels can be less."**

*Specific comments:*

*P2L12 : why using plural here? only O3 is a secondary air pollutant*

**The phrase has been changed to singular.**

*P5L14: the air pollution regimes (REC/SAD/NAD) should be introduced here and related to synoptic meteorological situations.*

**Basic information about these three regimes are provided now in the following manner:**

**"Under low-gradient synoptic conditions, the combination of the strong convective conditions and the blocking effect of the mountain ranges induces an important vertical development of the boundary layer and mesoscale recirculation. During the night, north-easterly winds prevail over the basin and, after dawn, the eastern slopes of the Guadarrama range are progressively warmed up causing a clockwise turning of wind to an E and S during the day finalising with an SW component in the late afternoon. The drainage flows at night-time re-establish the north-easterlies. These events are commonly referred as recirculation (REC) episodes. The presence of the Azores high or low pressure systems over the Atlantic in front of the Iberian Peninsula generate advection of Atlantic air masses from the north (we will refer to these as Northern advective or NAD events) or from the south (Southern advective or SAD events)."**

*P6L10: indicate the range explored in terms of O3 dry deposition velocities. Was this impact assessed on the basis of free-tropospheric total ozone burden as in Stevenson et al. or rather surface ozone?*

**We performed runs multiplying the standard dry deposition velocity (calculated by the resistance approach as detailed in Manders et al., 2016) by a factor of 1.25 and 0.75. Comparisons of modelled surface O3 concentrations revealed minimal effect of this parameter on the results for the specific period. The total O3 burden was not examined since we were just were studying specific summer events in a limited region and the background concentrations were not expected to vary significantly for the aforementioned changes in O3 deposition velocity.**

**On this respect, we have included the following sentence in section 2.3:**

**"Initial sensitivity studies were performed with the base configuration (configured similar to the operational forecasts that are part of the CAMS regional ensemble as presented in Marécal et al. (2015)) to test the response of the model to changes in the deposition velocity of O3 because night-time dry deposition has been suggested as a factor that could strongly influence the ability of CTMs to simulate tropospheric O3 (Stevenson et al., 2006; Monks et al., 2015). The standard dry deposition velocity, calculated by the resistance approach (Manders et al., 016), has been multiplied by either a factor 1.25 or 0.75. The results (not shown here) reflected a minimal effect of this parameter on O3 concentrations in the chosen domain and period, and therefore deepening in this direction was discarded."**

*P7L2: what is the reference year for emissions used?*

**The reference year for the MACC-III emissions was 2011. This has been indicated in the text.**

*P9L5: the worsening of correlations when increasing resolution has been documented as "double penalty" in the field of meteorological forecast. It would be worth discussing in more detail the issue here*

**A brief description of the "double penalty" concept is now provided:**

**"This is known in meteorological modelling as the Double Penalty issue (Mass et al., 2002) and occurs when evaluating simulations using point observations. The high resolution runs may be penalized twice, for not capturing the occurrence of the event and also for not predicting the right location of the event while a low resolution simulation can only fail predicting the event."**

*P12L2: it is indeed frustrating that NO2 is not included in the detailed validation. It would also be interesting to see some basing meteorological validation, not clear why WRF outperforms IFS and there could be compensation of errors.*

**A table showing the Pearson's correlation factor (r) correlation factor for NO2 and NOx was also composed and it is shown below. However, it is fair to indicate that, since O3 was our target pollutant, traffic stations were excluded from the set of stations used for validation so the picture may be a biased. Moreover, the results for the ECMWF_HR_70 were not calculated. As shown in that table, NO2 correlates worse with observations than O3 as generally occurs in CTM's due the high variability. However, it is true that ECMWF runs correlate better than WRF runs probably due to the coarser resolution of the first with respect to the latter. This can be a good example of the "double penalty" situation. These are the results:**

| STATION | TYPE | $NO_2$ | | | | $NO_x$ | | | |
|---|---|---|---|---|---|---|---|---|---|
| | | ECMWF_5 | ECMWF_70 | WRF_5 | WRF_70 | ECMWF_5 | ECMWF_70 | WRF_5 | WRF_70 |
| VILLA DEL PRADO | RURAL | 0.481 | 0.576 | 0.42 | 0.437 | 0.454 | 0.535 | 0.348 | 0.371 |
| SAN MARTIN DE VALDEIGLESIAS | | 0.405 | 0.454 | 0.337 | 0.358 | 0.425 | 0.45 | 0.344 | 0.308 |
| EL ATAZAR | | 0.422 | 0.524 | 0.381 | 0.472 | 0.446 | 0.539 | 0.408 | 0.476 |
| SAN PABLO DE LOS MONTES | | 0.463 | 0.502 | 0.295 | 0.423 | 0.451 | 0.507 | 0.304 | 0.455 |
| CAMPISABALOS | | 0.328 | 0.419 | 0.333 | 0.36 | 0.31 | 0.418 | 0.314 | 0.332 |
| GUADALIX DE LA SIERRA | | 0.509 | 0.479 | 0.57 | 0.408 | 0.49 | 0.46 | 0.583 | 0.378 |
| ORUSCO DE TAJUNA | | 0.481 | 0.471 | 0.39 | 0.388 | 0.156 | 0.288 | 0.088 | 0.046 |
| ALCORCON | URBAN | 0.369 | 0.466 | 0.407 | 0.447 | 0.329 | 0.422 | 0.342 | 0.353 |
| TOLEDO2 | | 0.318 | 0.33 | 0.244 | 0.233 | 0.274 | 0.281 | 0.221 | 0.196 |
| ENSANCHE DE VALLECAS | | 0.483 | 0.474 | 0.266 | 0.372 | 0.421 | 0.459 | 0.217 | 0.268 |
| VILLAVERDE | | 0.436 | 0.437 | 0.322 | 0.333 | 0.348 | 0.412 | 0.224 | 0.229 |
| ARTURO SORIA | | 0.422 | 0.522 | 0.319 | 0.342 | 0.391 | 0.526 | 0.328 | 0.391 |
| FAROLILLO | | 0.36 | 0.427 | 0.29 | 0.371 | 0.298 | 0.446 | 0.217 | 0.409 |
| PLAZA DEL CARMEN | | 0.219 | 0.307 | 0.184 | 0.204 | 0.162 | 0.304 | 0.131 | 0.263 |
| GUADALAJARA | | 0.359 | 0.394 | 0.271 | 0.235 | 0.363 | 0.399 | 0.256 | 0.221 |
| MOSTOLES | | 0.404 | 0.507 | 0.44 | 0.401 | 0.363 | 0.48 | 0.382 | 0.319 |
| ARANJUEZ | | 0.343 | 0.283 | 0.204 | 0.211 | 0.322 | 0.239 | 0.168 | 0.159 |
| RETIRO | | 0.339 | 0.523 | 0.232 | 0.455 | 0.346 | 0.575 | 0.266 | 0.519 |
| TRES OLIVOS | | 0.472 | 0.55 | 0.378 | 0.476 | 0.529 | 0.593 | 0.39 | 0.495 |
| AZUQUECA DE HENARES | | 0.448 | 0.425 | 0.318 | 0.26 | 0.434 | 0.403 | 0.291 | 0.256 |
| BARAJAS - PUEBLO | | 0.595 | 0.505 | 0.213 | 0.416 | 0.567 | 0.501 | 0.176 | 0.323 |
| RIVAS-VACIAMADRID | SUBURBAN | 0.446 | 0.348 | 0.322 | 0.301 | 0.386 | 0.363 | 0.27 | 0.224 |
| AVILA 2 | | 0.502 | 0.487 | 0.499 | 0.467 | 0.482 | 0.471 | 0.5 | 0.476 |
| JUAN CARLOS I | | 0.482 | 0.467 | 0.28 | 0.328 | 0.469 | 0.458 | 0.257 | 0.314 |
| EL PARDO | | 0.493 | 0.433 | 0.483 | 0.473 | 0.509 | 0.449 | 0.505 | 0.49 |
| ALGETE | | 0.413 | 0.451 | 0.328 | 0.267 | 0.419 | 0.456 | 0.314 | 0.219 |
| MAJADAHONDA | | 0.301 | 0.423 | 0.352 | 0.341 | 0.293 | 0.434 | 0.349 | 0.31 |
| ESTACION DE LA SAGRA (ILLESCAS) | | 0.322 | 0.344 | 0.274 | 0.198 | 0.282 | 0.29 | 0.232 | 0.134 |
| TORREJON DE ARDOZ | | 0.521 | 0.488 | 0.466 | 0.407 | 0.503 | 0.493 | 0.453 | 0.338 |
| VALDEMORO | | 0.449 | 0.422 | 0.385 | 0.378 | 0.422 | 0.437 | 0.406 | 0.327 |
| CASA DE CAMPO | | 0.243 | 0.378 | 0.264 | 0.39 | 0.183 | 0.391 | 0.177 | 0.423 |

**Regarding, the evaluation of the meteorological datasets we made a qualitative validation of the degree of concordance between the two meteorological datasets (WRF and ECMWF) and observations (at El Retiro in the centre of Madrid) of temperature, wind direction and wind speed. We concluded that temperature compared almost optimally for both datasets.**

Regarding the wind direction was is adequately represented in the two meteorological datasets. Finally, wind speed is occasionally overestimated by the meteorological models although this overestimation is more marked in the case of WRF (see below the plots of the comparison with WRF data were a comparison between the BLH provided by the input meteorological data and the LOTOS-EUROS transformation has been added).

With respect to the validation of meteorological fields in the vertical, Figure S4 in the supplementary information contains the comparison between the vertical fields of RH, T, and wind speed and direction.

[Figure]

*P16L25: how do you explain the local minima of O3 around 4km agl?*

We do not have a clear answer to this question. However, an analysis of the vertical profiles and concentration maps revealed that the band with high O3 located just below this local minima may correspond to an intrusion of O3 generated southeast from the MAB.

*P18, Fig 7: what is driving the sharp horizontal convergence of NO2 between 12 and 18UT?*

The convergence of NO2 is probably driven by convection since temperature was very high during the described event. Moreover, the presence of the Guadarrama range also influences. Finally, the presence of the NO2 column in the plots in the centre of the domain is logically associated with the region of high NOx emissions (Madrid metropolitan area).

*P20L15-20: this section deserves further discussion in light of Querol et al. 2018 (Section 4), where they analyze the likelihood of impact at the ground of the STE event in relation with diurnal PBL variability.*

**We have done this. The following lines have been introduced in the paper:**

**"The actual impact of this stratospheric intrusion on surface levels remains unclear. In the referenced paper, the authors estimate a possible but limited impact of the intrusion on surface levels assuming that the boundary layer could exceed the 3000 ma.s.l. during the day. As shown in Figure 9, LOTOS-EUROS predicts that the maximum altitude of the boundary layer according to LOTOS-EUROS reached its maximum values of 2500–2700 ma.s.l. limited by the wind ventilation so, probably, the impact on the surface should be low (if any) in this case."**

**This is true for 18/7. Thus, this sentence has been included in the text for clarity.**

*Figure 6, 8, 10: those figures are very nice and comprehensive. The only missing information is modelled O3 time series. Although model and observations are compared in the in 3.1.1 and 3.1.2, I am missing a visual comparison of time series. The vertical cross sections are difficult to read and would benefit from being consistent with Fig 7&9 (i.e. both could extend to 7km to allow discussing the stratospheric intrusion). The color legend of O3 for the vertical cross section should be consistent with the maps (and the label corrected: it is probably ppb, not ppm).*

**In these figures our main interest was the environmental interpretation of the O3 episodes with the valuable information provided by the high resolution simulation. This, along with the need of keeping the figure readable, advised against including the O3 time series for comparison especially, taking into account that a complete evaluation of the model performance has been already provided in the first sections.**

**Nevertheless, in correspondence with the interest of the reviewer, we have added a figure in the supplementary information. In that figure we show the modelled (WRF_70) vs observations time series of July 2016 in four stations representative of the different areas of the MAB: ALG on the central part of the MAB, AZU on the Henares valley at the northeast, ORT on the east and SMV on the southwest.**

**The objective of Figures 7, 9 and 11 was to show the behaviour of the vertical behaviour of NO2 and O3 in the relevant part of the troposphere (first 5 km) with good resolution. Meanwhile, the objective of the time cross-sections in Figures 6, 8 and 10 was to analyse the time evolution of concentrations so extending the vertical dimension up to 14 km allowed to observe the formation and evolution of, for example, stratospheric intrusions. This is the reason why we have chosen different vertical extent in the cross- sections presented in these two sets of figures. With respect to the different colour legends between the maps and the cross sections was compulsory since the concentration ranges were different in both cases. Finally, we have corrected the label indicated by the reviewer.**

---

## Author Comment (AC2) · 1 Oct 2019

*Response to reviewer #3 of "Analysis of summer O3 in the Madrid air basin with the LOTOS-EUROS chemical transport model" by Miguel Escudero et al.*

**The authors would like to acknowledge the comments from reviewer #3. We thank the positive comments. All the suggestions demonstrate a good knowledge of the scientific field and that has resulted in a great improvement of the paper after his/her revision.**

**Before answering to the reviewer's comments one by one, the authors want to clarify that the main objective of this paper is to perform a detailed analysis of summer O3 events in a topographically complex basin such as the MAB with the valuable help of LOTOS-EUROS CTM. This model was adapted/configured to provide the best representation of the scientific observations resulting from a comprehensive field campaign carried out in Madrid on July 2006. Our main goal was not to realise a deep analysis of the driving factors of LOTOS-EUROS in O3 simulations (physical and chemical schemes, emission patterns…).**

**We opted to focus our efforts in finding the best configuration for the particular study area and period. In addition to the horizontal resolution, we knew (from previous experimental studies in the region) that the vertical resolution was a key aspect that had not been taken into account so often in previous simulation studies in the Mediterranean. Moreover, the use of two different meteorological input datasets was planned in order to evaluate if there was a relevant difference in the outputs in the two cases. That is a relevant question for CTMs especially when modelling O3.**

What is the state of the art terms of fine restitution of the fields of ozone in the Mediterranean? What has been undertaken at this scale so far? What are the current shortcomings in terms of model performance for ozone, what are the locks for the restitution of fine-scale ozone fields, on the horizontal and on the vertical?

**The following paragraphs have been added to the introduction:**

**"Air quality model results vary at different resolutions especially due to the resolution of emissions and the description of the driving meteorology (Fenech et al, 2008). Some authors have found that the impact of higher horizontal resolutions in O3 simulations is more sensitive to the resolution of emissions than to meteorology (Valari and Menut, 2008). Moreover, finer resolution result in less dilution of emissions but also in differences have been found in the O3-NOx interaction (Valari and Menut, 2008, Stock et al., 2014, Schaap et al., 2015).**

**In the Iberian Peninsula, the use of fine grids (in the order of 1-5 x 1-5 km) has been found beneficial in the context of complex terrains where mesoscale processes acquire importance for interpreting production and transport of O3 (Toll and Baldasano, 2000, Jimenez-Guerrero et al., 2008,). In coastal areas, with complex topography, high resolution simulations have been generally employed with good results (Carvalho et al., 2006, Jimenez et al., 2006, Gonçalves et al., 2009). Moreover fine grids have been recommended for describing O3 variability especially in urban and industrial areas (Jimenez et al, 2006, Baldasano et al., 2011). In general, the use of finer resolution simulations in the Iberian Peninsula generally imply benefits in the O3 description such as improvement in correlation and reduction in bias and errors (Jimenez et al., 2006).**

Less importance has been given to the vertical resolution mostly because the vertical O3 profile evaluation of CTM is difficult due to the lack of experimental vertical O3 data. In complex domains in the Iberian Peninsula the models may not reproduce O3 concentrations due to a poor representation of mesoscale flows and layering and accumulation of pollutants (Gonçalves et al., 2009). In general, it has been demonstrated that incrementing vertical resolution would help to resolve meteorological phenomena (Carvalho et al., 2006) and would also offer a more realistic vertical exchange between the boundary layer and the free troposphere (Jimenez et al., 2006)."

*What is required to go further in terms of ozone modelling?*

From the generic point of view, there are a variety of questions that could be named as next steps for the improvement of CTMs regarding O3 modelling. In general, simulating episodes is challenging due to the impact of multi-driving factors (emissions, meteorology, physical-chemical schemes…). Thus, investigating responses to emission changes (including biogenic emissions that are poorly described) is obviously one of the key aspects but it also relevant to find specific configurations for the CTMs in order to provide the best possible estimates of the formation and transport mechanisms that trigger O3 concentrations especially in summer. For these and other challenges, it is extremely important to evaluate modelling tools with data obtained in campaigns/measurements using state-of-the-art equipment both on the surface and vertically.

More in particular, on the basis of the results obtained in the present work we would like to mention four particular studies/strategies that could favour CTM development:

- Include vertical measurements of NOx in field campaigns.
- Perform measurements of NOy for carrying out NOx/VOC sensitivity studies (according to Sillman's methodology).
- Improve temporal resolution in emission models.
- Study the vertical exchanges of O3 in detail.

These arguments are now reflected in the paper in the conclusions section:

"The results from this study can be useful to understand the phenomenology of high O3 episodes in the MAB and to gain knowledge to design appropriate strategies for air-quality management. Further research must be implemented to investigate aspects like the sensitivity to emission reduction scenarios or the role of VOCs with emphasis on the biogenic ones. Moreover, to perform the tasks of validating and optimising CTMs, increasing efforts should be made to conduct more field campaigns in different air basins in the Mediterranean using state-of-the-art equipment to generate data and knowledge about O3 behaviour both on the surface and vertically. Useful parameters to be included in these campaigns are O3, NOx, VOC and, when possible, intermediate products like NOy, HNO3 and H2O2 that, according to previous experience (Sillman, 1995), are key parameters for facing model-based NOx–VOC sensitivity studies and the assessment of emission inventories. Some of these parameters (especially NOx) should also be incorporated in vertical measurements.

In future, similar simulations to the one presented in this study should be performed in the different air basins in the IP where O3 exceedances have been recorded (Querol et al., 2016). CTMs should be configured specifically for each region or air basin to assure the best performance by capturing the influence of topography and local circulations. For such studies,

we highlight the importance of conducting experimental campaigns that can support the necessary model evaluation.

Finally, it should be noted that when running such fine resolutions for real applications it is also important to work on the emission datasets (out of the scope of this work). Increasing the detail in emission inventory (mainly based on a bottom-up approach) could improve the performance of CTMs when assessing sensitivities or emission scenarios. Moreover, improving time resolution in the emission models can be beneficial for simulating O3 episodes."

*In what way does this fine-scale study bring new elements, not only to the evaluation of the model itself, but also to research on the subject?*

The topographical complexity of the MAB along with the large variability of emissions in a conurbation like Madrid required the use of fine scale configurations. In the vertical, it is clear that a considerable number of layers in the model allow a better representation of the vertical variability of O3 that would have been more difficult to observe and analyse with a coarser resolution (stratospheric intrusion, formation of residual layers, etc). This has been stated by many authors who carried out other fine-scale modelling studies (see discussion about fine resolution studies added to the introduction above). Moreover, the temporal extent of the analysis (1 month) is not common and has offered the opportunity to describe the patterns/scenarios of O3 episodes in the MAB during summer. Finally, it is innovative the discussion of the role of the BLH because this parameter is highly dependent of the model resolution.

*Also, why do the authors only focus on model resolution?*

In these detailed analysis for regions with complex topography and a great variety of sources, model resolution in the horizontal is an issue. Also vertical resolution was a key aspect on the light of previous studies which studied O3 phenomenology based on experimental campaigns (Querol et al., 2018 among others). In those studies, vertical fumigation was observed to have a key influence on surface O3 concentrations. In consequence, if LOTOS-EUROS was able to adequately represent mechanisms such as vertical exchange, formation of O3 residual layers, high altitude intrusions or fumigation according to what had been observed in the field campaigns, it could be trustfully used for a phenomenological study on O3 in summer in the MAB. We should bear in mind that LOTOS-EUROS in its standard configuration is a computationally cheap CTM (fast and accurate) and, by this study, we demonstrate that it is also able to reproduce complex events with an obvious resolution adaptation.

*In the conclusions, there is a lack of discussion about the importance of improvements (improved indicators), with regard to the requirements of enhanced configurations (especially CPU time).*

The question of whether it is worthwhile to perform these high resolution simulations (both horizontally and vertically) in terms of CPU time depends on the objectives that you are seeking. In this case, CPU time was increased reasonably when running 70-layer schemes with fine resolution (ECMWF_70 or WRF_70) because our objective was to describe O3 phenomenology in the MAB.

**We have added to the conclusions section a discussion about this:**

"**The main objective of the paper is to provide a phenomenological interpretation of O3 events in the area after performing a detailed evaluation of the best configuration of the model for the specific area and period. Regarding the specific question of the reasonable number of vertical levels in the model configuration, it is dependent on the objective of the study. In this study the environmental analysis was the main objective and it was logical and feasible from the perspective of CPU time to employ a considerable number of vertical levels because it allowed a better representation of the vertical variability of O3. In other studies such as air quality forecasting or long term analyses in which CPU time may be large, the reasonable number of levels can be less.**"

*It would also be important to give a strategy to choose a future configuration: is it reasonable to choose so many vertical levels (70)?*

**This answer is linked with the previous one. The use of 70 layers allows the model to represent a vertical complexity that would not be possible with the usual number of layers in standard CTM configurations. This was important for this work because we aimed to perform a correct phenomenological study of O3 in a complex region. Is it reasonable to make operational analyses or forecasts with that high number of layers? Probably we cannot afford that but the objective is different in those cases. Just to highlight the relevance of the vertical configuration of the model for O3 simulations, this study has triggered a discussion among the developers of LOTOS-EUROS on incrementing the number of vertical layers in the standard version of the model without putting in risk the computational efficiency of the model. As commented in the previous question, a paragraph discussing about this has been added to the conclusions section.**

*Also, still for a contextualization (this time of the results and not of the stakes), the interpretations of the episodes (which are precise), lack some context. There has been indeed a lot of studies of ozone formation episodes in the literature. The formation of ozone downwind sources, and its dependence upon wind speed and vertical dilution are known: what exactly are the new knowledge at the end of the study, concerning the phenomenology of episodes and their properties, or concerning the ability of a CTM to simulate them? How can these new knowledge improve air quality management strategies?*

**This study provides new knowledge on the occurrence of O3 episodes in the MAB. Among these, we could highlight three. Firstly, it is the first time that summer O3 events in the MAB have been classified in detail, taking into account transport features and the relevance of vertical exchange. This is then a necessary step to investigate emission scenarios and control strategies to be applied. Moreover, this study is innovative because it demonstrates the transfer of O3 produced at the MAB towards other air basins under certain meteorological scenarios (see, for example, section 3.2.3). Finally, regarding the ability of CTM to simulate complex O3 events in Southern Europe, this study demonstrate the relevance of horizontal and, especially, vertical resolution in CTM configuration.**

*Are we just talking about improved scores from one version to another, or do some configurations allow to depict specific features that do not appear in the others?*

**Apart from the relevant improvement of scores associated with increments in the horizontal resolution, it is clear that, for example, the 5-layer schemes (ECMWF_5 and WRF_5) cannot describe vertical exchange of O3 since they do not have enough resolution. This means that, with low number of layers, it is impossible, for example, to observe the formation of residual layers or the influence of stratospheric intrusions.**

*Specific comments:*

*Page 10 – line 5 - "In the plots corresponding to ECMWF_5 and ECMWF_70 runs we observe systematic positive bias especially in the period 14–20 UTC when the formation is strong although it only spiked with low wind speed. This feature was not so marked in the three remaining configurations and, in particular, in the two WRF runs the bias values were randomly distributed around zero." How do the authors explain the mid-day biases of ECMWF compared to WRF? Is it just a problem of resolution of the meteorological calculations, or does it depend on the meteorological model itself?*

**We suspect that it can be attributed mainly to the resolution given that the ECMWF run performed with higher resolution (ECMWF_HR_70) responded more closely to the WFR runs.**

*Line 15 page 15: the best overall performance is analyzed on which criteria? Which parameters are used to affirm that the simulation is better? Should it be the restitution of the diurnal peak, the phasing of the morning increase, the total amount on the vertical, or just the indicator average...? In particular, the WRF70 has a strong underestimation of diurnal ozone in Figure 4 at El Pardo and this is still considered as the "best run". What about this feature at other stations?*

**The best overall performance was selected from all parameters obtained in the validation assessment:**

   i.    **From the inspection of r values we concluded that the configurations with higher vertical resolution perform better than the 5-layer schemes.**
   ii.    **From the analysis of the model bias (Figures 2, 3 and S3), we concluded that clear improvements were observed using finer spatial resolution so either WRF simulations (WRF_5 and WRF_70) or ECMWF_HR_70 were the best option on this regard.**
   iii.    **The assessment of NMSE (Figure 2) revealed that WRF_70 showed slightly lower errors than the other configurations although differences were not drastic.**
   iv.    **After the evaluation of mean daily cycles (Figure 4) we found that the five-level configurations (ECMWF_5 and WRF_5) presented a not realistic sharp increment in O3 concentrations from 6 to 7 UTC. Although the timing of the increase was better reproduced in these runs than in the 70-level schemes, the latter were more realistic in the daily evolution of O3 concentrations (without steep increases in the morning).**
   v.    **From the comparison of vertical profiles, we concluded that obviously high resolution in the vertical was needed to capture the layering of pollutants reflected in the O3 soundings.**

**Summarising, the best overall configuration for the analysis of O3 in this region/period were setups with high resolution both in the vertical and in the horizontal, namely ECMWF_HR_70**

**and WRF_70. In particular, we used WRF_70 for vertical cuts because spatial resolution over the study area was better.**

**The underestimation of diurnal O3 in the mean daily cycle at El Pardo was not present in other stations on a routine basis. The following pictures present cases in which the mentioned underestimation is not observed.**

[Figure]

*The difference between what is observed and simulated is not always specified, even if we can guess it. Example: in "Figure 7. Longitudinal and latitudinal vertical crosssections of NO2 and O3 for 16 July 2016". Same for figure 9 and figure 11. Also, for page 17 lines 1 to 10, it is important to specify that it is a vision drawn by the model and not the result of observations.*

**We have made this clear in those Figures' captions. In page 17 we have added a comment making clear that the interpretation is made by means of the analysis of simulation results:**

**"From the simulation results, we can interpret the evolution of O3. At 18 UTC of 16 July, we can see how O3 levels increased drastically, and the boundary layer depth grew up to 3500 m a.s.l. aided by convection at 18 UTC. Normally, REC events show higher planetary boundary layer (PBL) heights in the evening. Figure 7 shows how the strong convection during REC events injected ground-level pollutants at high altitudes during the late afternoon 5 and the evening reaching up to 3500 m a.s.l. as illustrated in the NO2 plots. When the night-time stable boundary layer forms after sunset, air masses with high O3 that originated near the surface during the previous day were decoupled and remained in the residual layer at altitudes ranging between 2000 and 4000 m a.s.l. forming reservoir layers (00 and 06 UTC cross-sections in Figure 7) which can fumigate the following day. These reservoir layers can also be observed as a relatively thin band at an altitude of 2000–4000 m a.s.l. during every night of the REC period (Figures 6 and S5)."**

*Figures 6, 8, 10: The location/typology of the station groups should be mentioned.*

**This information is presented in Figure 1. Thus, in order to clarify we have added in the figure caption of Figures 6, 8, 10 the reference to Figure 1.**

*Figure 7: it would be more convenient for the reader to visualize on a map the latitudinal and longitudinal cuts.*

**Exactly as in the previous comment, the map with the latitudinal and longitudinal cuts is presented in Figure 1. We refer again in the figures' captions to Figure 1.**